# MAP Estimation with Denoisers: Convergence Rates and Guarantees

**Scott Pesme**     **Giacomo Meanti**     **Michael Arbel**     **Julien Mairal**

Univ. Grenoble Alpes, Inria, CNRS, Grenoble INP, LJK

`firstname.lastname@inria.fr`

## Abstract

Denoiser models have become powerful tools for inverse problems, enabling the use of pretrained networks to approximate the score of a smoothed prior distribution. These models are often used in heuristic iterative schemes aimed at solving Maximum a Posteriori (MAP) optimisation problems, where the proximal operator of the negative log-prior plays a central role. In practice, this operator is intractable, and practitioners plug in a pretrained denoiser as a surrogate—despite the lack of general theoretical justification for this substitution. In this work, we show that a simple algorithm, closely related to several used in practice, provably converges to the proximal operator under a log-concavity assumption on the prior $p$. We show that this algorithm can be interpreted as a gradient descent on smoothed proximal objectives. Our analysis thus provides a theoretical foundation for a class of empirically successful but previously heuristic methods.

## 1   Introduction

Inverse problems are ubiquitous in scientific and engineering fields involving image acquisition. In many such problems, the object of interest is not directly observed but instead undergoes a degradation process—such as blurring, downsampling, or noise corruption. The goal is to reverse this degradation and recover the original image.

A classical approach formulates this task as an optimisation problem balancing two terms: a *data fidelity term*, modelling the observation process, and a *regularisation term*, encoding prior knowledge about the solution. Historically, regularisers such as total variation or wavelet sparsity were hand-crafted [Mallat, 1999]. While effective to some extent, recent approaches often rely on *data-driven priors*, using pretrained denoisers and generative models. In particular diffusion and flow-based models offer powerful ways to learn the true image distribution $p$ from large datasets.

This opens the door to principled formulations like *Maximum a Posteriori (MAP)* estimation:

$$\arg \min_{x \in \mathbb{R}^d} \lambda f(x) - \ln p(x), \tag{MAP}$$

which corresponds to the posterior mode under the likelihood $p(y \mid x) \propto \exp(-\lambda f(x))$ and prior $p(x)$. In practice, however, this optimisation problem is extremely challenging to solve: evaluating the score $-\nabla \ln p(x)$ is often intractable, the term $-\ln p(x)$ can be severely ill-conditioned, and the data fidelity term $f(x)$ is frequently not strongly convex. A wide range of methods have been proposed to address these problems, and many of them perform remarkably well empirically. Yet, these methods do not come with the guarantee of actually minimising the MAP objective, making their success difficult to interpret.

A natural class of algorithms for addressing the MAP optimisation problem are proximal splitting methods [see, *e.g.*, Beck and Teboulle, 2009, Figueiredo et al., 2007, Combettes and Pesquet, 2011], which are particularly effective when dealing with objectives that combine smooth and non-smooth

39th Conference on Neural Information Processing Systems (NeurIPS 2025).

components. These methods alternate between two steps: one that follows the gradient of the data fidelity term, and another that incorporates prior knowledge through what is known as a "proximal update" — a correction step informed by the prior distribution.

However, for prior models relying on an unknown data distribution, this proximal update is extremely difficult to compute exactly. To circumvent this, a popular line of work introduced by Venkatakrishnan et al. [2013] known as Plug-and-Play (PnP) replaces the intractable proximal step with a pretrained denoising neural network. PnP methods have shown excellent empirical performance in a wide range of inverse problems. But despite their success, they come with a significant caveat: the denoiser is not designed to match the proximal operator it replaces. As a result, the overall algorithm no longer corresponds to solving the original MAP estimation problem, which limits its interpretability and makes it hard to analyse theoretically unless strong constraints are imposed on the denoiser [Hurault et al., 2022, Sun et al., 2021, Hertrich et al., 2021, Cohen et al., 2021].

More recently, a new wave of approaches has emerged which view inverse problems as a sampling task, see [Delbracio and Milanfar, 2023, Chung et al., 2023, Boys et al., 2024] among others, moving further away from traditional optimisation frameworks. One example is the Cold Diffusion [Bansal et al., 2023] algorithm, which combines denoising steps with corruption steps towards the observed data, with decreasing intensity. While this method often produces high-quality results in practice, especially with a small number of steps, it also lacks strong convergence guarantees and may diverge during extended runs with default parameters [Delbracio and Milanfar, 2023].

In this work, we revisit denoising-based iterative schemes from a theoretical perspective, focusing on the case where the negative log-density $p$ is log-concave and potentially ill-conditioned. Specifically, we show that a simple algorithm originally proposed by Bansal et al. [2023] with appropriate step-sizes converges to the proximal operator of the negative log-density, and we establish corresponding convergence rates. Having a reliable approximation of the proximal operator enables its integration into broader MAP estimation frameworks, akin to Plug-and-Play methods, but now supported by a rigorous theoretical foundation.

**Our contribution: establishing convergence rates for MAP estimation.** In this work, for a suitable choice of sequences of noise levels $\sigma_k \geq 0$ and weights $\alpha_k \in (0, 1)$, we consider the following recursion to compute the proximal operator of $-\ln p$ at a point $y \in \mathbb{R}^d$:

$$x_{k+1} = (1 - \alpha_k)\,\mathrm{MMSE}_{\sigma_k}(x_k) + \alpha_k y, \qquad \text{(MMSE Averaging)}$$
$$\text{with} \quad \mathrm{MMSE}_\sigma(z) := \mathbb{E}[X \mid X + \sigma\varepsilon = z],$$

where the expectation is taken over $X \sim p$ and $\varepsilon \sim \mathcal{N}(0, I_d)$ conditionally on $X + \sigma\epsilon = z$. In practice, the theoretical minimum mean square error denoiser $\mathrm{MMSE}_\sigma$ can be approximated by a neural network which has been trained to match the MMSE denoiser.

Each iterate in the recursion is a weighted average between a denoised version of the current point and the original input $y$, echoing the structure of methods like Cold Diffusion [Bansal et al., 2023]. What makes this recursion striking is that, for appropriate choices of weights $\alpha_k$ and vanishing noise levels $\sigma_k \to 0$, it can be rewritten (see Proposition 1)—via the Tweedie formula [Efron, 2011]—as:

$$x_{k+1} = x_k - \alpha_k \nabla F_{\sigma_k}(x_k), \quad \text{with} \quad F_{\sigma_k}(x) := \frac{1}{2}\|y - x\|^2 - \tau \ln p_{\sigma_k}(x),$$

where $p_\sigma$ denotes the convolution of the prior $p$ with a Gaussian of variance $\sigma^2$. Under this reinterpretation, the recursion corresponds to gradient descent on a sequence of smoothed objectives $(F_{\sigma_k})_k$ converging to the true proximal objective $F(x) := \frac{1}{2}\|y - x\|^2 - \tau \ln p(x)$ whose minimiser is the proximal point $\mathrm{prox}_{-\tau \ln p}(y)$. This perspective enables a rigorous convergence analysis: as $\sigma_k \to 0$, each update more closely resembles a step on $F$, and the iterates can be shown to converge to its minimiser.

> We show that, under a log-concavity assumption on $p$ and a bound on the third derivative of $-\ln p$, the iterates $x_k$ of the MMSE Averaging recursion converge to the true proximal point at the following rate (see Theorem 1):
>
> $$\left\| x_k - \mathrm{prox}_{-\tau \ln p}(y) \right\| \leq \tilde{O}(1/k),$$
>
> where $\tilde{O}(\cdot)$ hides logarithmic factors. Importantly, our convergence bound does not rely on the $L$-smoothness constant of the negative log-prior $-\ln p$, which could be arbitrarily large.

This result provides theoretical grounding for algorithms that previously lacked a variational interpretation, establishing a direct connection between heuristic denoising schemes and principled optimisation algorithms. Crucially, it yields an explicit method to approximate the proximal operator of the negative log-prior—a central building block in many optimisation frameworks for inverse problems [Venkatakrishnan et al., 2013, Romano et al., 2017, Hurault et al., 2021]. Once available, this proximal operator can be readily integrated into broader algorithms, such as proximal gradient descent and its accelerated variants [Beck and Teboulle, 2009]. In Theorem 2, we demonstrate exactly this by plugging our approximation into a proximal gradient method to solve the MAP problem.

The proof of convergence with explicit rates of the MMSE Averaging iterates towards the proximal operator, while conceptually intuitive, requires a careful blend of inexact optimisation analysis and tools from partial differential equations—most notably the heat equation—to control how the minimiser of the smoothed objectives $F_\sigma$ evolves with the noise level.

## 2 Related Works

Our work shares similar motivations with much of the literature on Plug-and-Play (PnP) methods for inverse problems [Venkatakrishnan et al., 2013]. The PnP literature is vast, and for a particularly clear and comprehensive overview, we refer the reader to the PhD thesis of Samuel Hurault [Hurault, 2023]. PnP methods replace the proximal operator $\text{prox}_{-\tau \ln p}(y)$ with a generic denoiser $D_\sigma$, typically parameterised by the noise level $\sigma$. A wide variety of denoisers have been used, including classical approaches [Dabov et al., 2007, Zoran and Weiss, 2011], CNN-based denoisers [Zhang et al., 2021, Kamilov et al., 2023, Zhang et al., 2017] and, more recently, diffusion models [Graikos et al., 2022, Zhu et al., 2023]. These methods are often combined with different optimisation schemes (e.g., PGD [Terris et al., 2020], ADMM [Romano et al., 2017], HQS [Zhang et al., 2017]) and adapted to different specific inverse problems. Several works [Sreehari et al., 2016, Gavaskar and Chaudhury, 2020, Nair et al., 2021, Xu et al., 2020] show that a variety of PnP algorithms converge, however they cannot guarantee that the denoiser is a proximal operator, let alone the proximal operator of the correct functional. Furthermore the convergence proofs often rely on restrictive assumptions on the denoising model [Reehorst and Schniter, 2018]. Indeed, the denoiser is usually trained [Zhang et al., 2021, Meinhardt et al., 2017] to minimize the MSE and hence—under Gaussian noise assumptions—converges to the MMSE estimator which can be very different from the MAP [Gribonval, 2011].

Gradient step (GS) denoisers [Cohen et al., 2021, Hurault et al., 2021] parameterize $D_\sigma = I - \nabla g_\sigma$, where $g_\sigma$ is a neural network. It is then possible to show that $D_\sigma$ is indeed the proximal operator of an explicit functional [Hurault et al., 2022], but this function is unfortunately not the negative log prior as desired. Similarly, Hauptmann et al. [2024] link linear denoisers to the proximal operator of a regularization functional, which is however again not $-\ln p$.

Two recent theoretical works share our concerns about existing PnP methods and strive to learn the correct proximal operator: Fang et al. [2023] replace the usual MSE loss by a proximal matching loss which is guaranteed in the limit to yield $\text{prox}_{-\tau \ln p}$. Though elegant, they do not establish any convergence rate, and their training procedure only approximates the desired limit without providing a bound on the approximation error. Using an approach somewhat close to ours, Laumont et al. [2023] introduce PnP-SGD, which performs stochastic gradient descent on a smoothed version of the proximal objective $F_\sigma$. However, by keeping the smoothing parameter fixed ($\sigma_k = \sigma$), their method only converges to the proximal operator of the *smoothed* density and the convergence rate depends on the smoothness constant of $F_\sigma$, which can be arbitrarily large and lead to slow convergence as explained in this work.

The second class of approaches which are receiving more and more attention in the context of solving inverse problems are conditional diffusion methods. These algorithms are typically based on modifying the smoothed prior score $\nabla \ln p_\sigma(x_\sigma)$—obtained through a pretrained diffusion model— into the posterior score $\nabla \ln p_\sigma(x_\sigma \mid y)$. Coupled with sampling along the reverse diffusion SDE this allows to generate samples from the desired probability distribution. Dhariwal and Nichol [2021] propose to use a classifier to estimate $\nabla \ln p(y \mid x)$, Jalal et al. [2021] approximate $p_\sigma(y \mid x_\sigma) \approx p(y \mid x)$ obtained through the explicit likelihood term under Gaussian noise, the DPS algorithm [Chung et al., 2023] approximates the mean of the smoothed log prior with the Tweedie formula and Boys et al. [2024] additionally approximates the standard deviation. All such methods aim to sample from the posterior distribution rather than identify its maximum. Moreover, they rely on approximations that are difficult to control, offering no guarantees of sampling from the true posterior. Although

asymptotic guarantees can be achieved with more sophisticated algorithms [Wu et al., 2023], these methods are not designed to recover the MAP estimate.

Using flow matching, Zhang et al. [2024] approximate the MAP solution directly, without relying on the proximal operator. Instead, they construct a trajectory that trades off between the prior and data fidelity terms, but no convergence rates are given. Finally, Ben-Hamu et al. [2024] solve a similar problem, but additionally need an expensive backpropagation step through an ODE at every step.

# 3  Main Result: Convergence Towards the Proximal Operator

We begin by showing that the MMSE Averaging recursion corresponds to gradient descent on a sequence of smoothed approximations of the proximal objective $F$. We then show that these smoothed objectives are significantly better conditioned than the original unsmoothed problem. Finally, we prove convergence of the iterates and provide explicit convergence rates.

## 3.1  From MMSE Averaging to Gradient Descent on Smoothed Proximal Objectives

We can connect the recursion in MMSE Averaging to the negative log-prior $-\ln p$ by leveraging the celebrated Tweedie identity (see for example Efron [2011]), which links the MMSE denoiser to the gradient of the log-density of a smoothed version of the prior. Specifically, if $p_\sigma$ denotes the Gaussian convolution of $p$ with a centred Gaussian of variance $\sigma^2$ (i.e. the density of $X + \sigma\varepsilon$), then:

$$\text{MMSE}_\sigma(z) = z + \sigma^2 \nabla \ln p_\sigma(z).$$

Plugging the above identity into the MMSE Averaging recursion allows expressing the iterate update in terms of the score of the smoothed density $p_{\sigma_k}$, which already resembles a gradient descent update:

$$x_{k+1} = x_k - \alpha_k \left( (x_k - y) - \frac{(1 - \alpha_k)}{\alpha_k} \sigma_k^2 \nabla \ln p_{\sigma_k}(x_k) \right).$$

Rearranging the terms in the above expression naturally leads to the following simple observation:

**Proposition 1.** *The MMSE Averaging recursion with choice of weights $\alpha_k = 1/(k + 2)$ and noise sequence $\sigma_k^2 = \tau/(k + 1)$ can be rewritten:*

$$x_{k+1} = x_k - \alpha_k \nabla F_{\sigma_k}(x_k), \quad \text{with} \quad F_{\sigma_k}(x) := \frac{1}{2}\|y - x\|^2 - \tau \ln p_{\sigma_k}(x).$$

In this form, the recursion is naturally interpreted as a gradient descent algorithm applied to a sequence of smoothed proximal objectives $(F_{\sigma_k})_k$ and with stepsizes $(\alpha_k)_k$. This reformulation not only enables a clean convergence analysis but also offers a new perspective on the MMSE Averaging recursion: as $\sigma_k \to 0$, one can hope that the iterates approach the minimiser of the original (unsmoothed) proximal objective:

$$F(x) := \frac{1}{2}\|y - x\|^2 - \tau \ln p(x). \qquad \text{(Proximal Objective)}$$

Moreover, we argue that this smoothed approach leads to faster convergence than applying standard gradient descent directly to the original, potentially badly conditioned Proximal Objective.

## 3.2  Good Conditioning Properties of $F_\sigma$

Compared to the original objective $F$, the function $F_\sigma$ enjoys much better properties. In particular, the next result shows that $F_\sigma$ is $L_\sigma$-smooth, with smoothness controlled by the noise level $\sigma$.

**Proposition 2.** *For any $\sigma > 0$, the function $F_\sigma$ is $L_\sigma$-smooth, with*

$$L_\sigma = 1 + \frac{\tau}{\sigma^2}.$$

The proof can be found in Section A and is a simple consequence of known results on the Hessian of $-\ln p_\sigma$. This smoothing effect introduces a natural trade-off: for large $\sigma$, the objective $F_\sigma$ becomes easier to minimise thanks to an improved smoothness, but the minimiser of $F_\sigma$ may deviate significantly from that of the original problem. While this smoothness property holds for any density

function $p$, obtaining convergence guarantees requires stronger assumptions. In particular, we will focus on the case where $p$ is log-concave and satisfies regularity conditions. Although this assumption is clearly idealised and does not hold for many practical distributions, it offers a manageable setting for theoretical analysis.

**Assumption 1.** *The density $p$ is log-concave, and strictly positive on $\mathbb{R}^d$.*

In particular, this ensures that $-\ln p$ is well-defined and convex over $\mathbb{R}^d$, so that the Proximal Objective function $F$ is 1-strongly convex and admits a unique minimiser, denoted by $\text{prox}_{-\tau \ln p}(y) := \arg\min F$. Furthermore, the stability of log-concavity under convolution (a special case of the Prékopa–Leindler inequality, see [Saumard and Wellner, 2014, Proposition 3.5.]) ensures that $-\ln p_\sigma$ is convex for all $\sigma > 0$, and hence that $F_\sigma$ is 1-strongly convex. Along with Proposition 2, this allows to quantify how much the smoothing improves the conditioning of the objective in the following proposition.

**Proposition 3.** *Under Assumption 1, for $\sigma > 0$, the function $F_\sigma$ is $L_\sigma$-smooth and $\mu_\sigma$-strongly convex with $L_\sigma = 1 + \tau/\sigma^2$ and $\mu_\sigma = 1$. The condition number of $F_\sigma$ is therefore at most*

$$\kappa_\sigma = \frac{L_\sigma}{\mu_\sigma} = \left(1 + \frac{\tau}{\sigma^2}\right).$$

This result highlights a key benefit of the smoothed proximal objective: as $\sigma$ increases the function $F_\sigma$ becomes significantly better conditioned, with the condition number $\kappa_\sigma$ decreasing toward 1 as $\sigma \to \infty$. For example, setting $\sigma = \sqrt{\tau}$ already yields a condition number of $\kappa_{\sqrt{\tau}} = 2$.

Next, we impose an assumption on the third derivative of the log-prior, which is crucial in our analysis for controlling the Lipschitz continuity of the map $\sigma^2 \mapsto \arg\min F_\sigma$. Without such control, it would be difficult to establish any meaningful convergence guarantees for the iterates of MMSE Averaging.

**Assumption 2.** *The prior $p$ is three times differentiable and the third derivative of $\ln p$ is bounded. We denote by $M \geq 0$ the quantity:*

$$\sup_{x \in \mathbb{R}^d} \left\| \nabla^3 \ln p(x) \right\|_F = M,$$

*where for $A \in \mathbb{R}^{d \times d \times d}$, $\|A\|_F = \left( \sum_{i,j,k} A_{ijk}^2 \right)^{1/2}$ corresponds to the Frobenius norm.*

This assumption controls how *skewed* and *"non-quadratic"* the log-prior is, and we make it in order to control the stability of the minimisers $\text{prox}_{-\tau \ln p_\sigma}(y) := \arg\min F_\sigma$ as $\sigma$ varies. Also note that an upper bound on the third derivative does not imply an upper bound on the second one: indeed for a Gaussian prior $p$, its negative log likelihood is a simple quadratic, which can have arbitrarily large $L$-smoothness, while its third derivative is trivially 0.

### 3.3 Convergence of the MMSE Averaging Iterates Towards the Proximal Operator

Leveraging the upper bound on the condition number of the objectives $(F_\sigma)_{\sigma \geq 0}$, we obtain the following convergence result on the iterates $x_k$ of the MMSE Averaging recursion:

**Theorem 1** (Convergence to the Proximal operator). *Under Assumptions 1 and 2, let $\text{prox}_{-\tau \ln p}(y)$ denote the unique solution of the Proximal Objective problem. Then, the MMSE Averaging iterates with parameters $\alpha_k = 1/(k+2)$, $\sigma_k^2 = \tau/(k+1)$ and initialised at $x_0 = y$ satisfy:*

$$\|x_k - \text{prox}_{-\tau \ln p}(y)\| \leq \frac{(\ln k) + 7}{k+1}\left[\|y - \text{prox}_{-\tau \ln p}(y)\| + \tau^2 M \sqrt{d}\right].$$

**Comparison with naive GD: illustration with a Gaussian prior.** The most important part of our result is that the convergence bound does not depend on the $L$-smoothness of $-\ln p$, which could be arbitrarily large. The convergence rate depends only on a bound on the *third* derivative of $-\ln p$, which may remain moderate even when the second derivative is large. This is unlike gradient descent (GD) applied directly to the proximal objective $F$, whose rate scales poorly with the $L$-smoothness of $-\ln p$. We illustrate this with a toy yet instructive case of a Gaussian prior, for which the third derivative of the log likelihood is trivially zero, yet the second derivative can be arbitrarily large. Let $p$ be the density of a $d$-dimensional Gaussian $\mathcal{N}(0, H^{-1})$, with $H$ a positive definite matrix whose smallest eigenvalue we arbitrarily consider to be $\mu = 1$ and whose largest eigenvalue $L \gg 1$ can be

arbitrarily large. In this setting the negative log-prior $-\ln p$ is a quadratic with Hessian $H$ and $F$ is a quadratic too with Hessian equal to $(I + \tau H)$. The corresponding smoothness constant of $F$ is therefore $L_F = 1 + \tau L$, and the strong convexity constant is $\mu_F = 1 + \tau$. Since $L_F$ can be arbitrarily large, gradient descent on $F$ requires an arbitrarily small (and non-practical) step size $\alpha < 1/L_F$. For $\alpha = 1/L_F$, the iterates satisfy the standard convergence bound:

$$\|x_k - \text{prox}_{-\tau \ln p}(y)\| \leq \left(1 - \frac{\mu_F}{L_F}\right)^{k/2} \|y - \text{prox}_{-\tau \ln p}(y)\|,$$

leading to an iteration complexity of $L \cdot \log(1/\varepsilon)$ to reach $\varepsilon$-accuracy. From Theorem 1, since $M = 0$ the MMSE Averaging iteration converges much faster, with rate $\tilde{O}(1/k)$ (i.e. iteration complexity $O(1/\varepsilon)$), which is tight up to the log term (see Section A.3).

**Parameter-free algorithm.** A key practical advantage of our result is that it guarantees convergence for a parameter-free choice of weights $\alpha_k$ and noise levels $\sigma_k$. Specifically, these sequences depend only on the chosen regularisation parameter $\tau$ and do not require any knowledge of smoothness or Lipschitz constants, condition number, or other problem-specific properties of the prior distribution $p$. This makes the algorithm particularly simple to use and eliminates the need for costly hyperparameter tuning.

**Sketch of proof.** The proof (given in Section A.3) combines techniques for approximate gradient optimization and a priori estimates on the solution to a partial differential equation. We begin by applying the standard descent lemma to the smoothed objective $F_{\sigma_k}$, which yields a contraction towards its minimiser at a rate determined by the condition number $\kappa_{\sigma_k}$ which is controlled through Proposition 3, guaranteeing consistent progress. However, because the minimiser of $F_{\sigma_k}$ changes with $\sigma_k$, we must control how much it drifts over the iterations. To do this, we study the evolution of the minimiser of $F_\sigma$ as a function of $\sigma$ by analysing the differential equation it satisfies. This is made possible by the fact that $p_\sigma$ satisfies the heat equation. The resulting ODE for $\arg\min F_\sigma$ involves the quantity $\nabla \Delta \ln p_\sigma$, which we are able to bound uniformly in $\sigma$ by $M\sqrt{d}$ by carefully analysing the parabolic inequality satisfied by $\|\nabla^3 \ln p_\sigma(x)\|_F$ and using the bound from Assumption 2 for $\sigma = 0$. Summing the incremental drift contributions and combining them with the contraction bound yields the final convergence result toward the true proximal point.

**Link with cold diffusion.** There is a notable similarity between our algorithm and a heuristic approach introduced in Bansal et al. [2023], which generates images by inverting a known degradation. When the degradation operator is defined as a linear interpolation between the degraded image $y$ and the clean image $x$ (as explained in Section 6.2 in Delbracio and Milanfar [2023]), cold diffusion initialises at $x_0 = y$ and applies the following iteration for a fixed number of steps $N$:

$$x_{k+1} = (1 - \alpha_k)D_\theta(x_k, k) + \alpha_k y, \quad \text{with } \alpha_k = 1 - \frac{k}{N}$$

where $D_\theta$ is a trained denoiser, as for our recursion MMSE Averaging. However, note that the choice $\alpha_k := k/N$ differs from the schedule used in our theoretical analysis. While this empirical scheme yields strong results for very small $N$, it lacks convergence guarantees and tends to diverge as the number of iterations increases. We suspect that this instability may be due to the fact that the fixed ratio $k/N$ does not necessarily correspond to a well-behaved weighting policy.

**Comparison with standard random smoothing techniques.** The smoothing that appears through $-\ln p_\sigma$ differs significantly from classical random smoothing approaches (e.g., Nesterov and Spokoiny [2017]). In standard random smoothing, the goal is to regularise a possibly non-smooth function $h$ by convolving it with a Gaussian, yielding a smooth approximation $h_\sigma(z) := \mathbb{E}_{\varepsilon \sim \mathcal{N}(0, \sigma^2 I)}[h(z + \varepsilon)]$. This smoothed function $h_\sigma$ inherits favourable differentiability properties that are well understood and can be leveraged in zeroth-order or gradient-based optimisation. In contrast, our approach considers the *logarithm of a smoothed function*—specifically, $-\ln p_\sigma$, where $p_\sigma$ is the Gaussian convolution of a density $p$. This subtle change has a major impact: the logarithm does not commute with convolution, and the resulting function exhibits different analytic properties. As a result, existing results from the standard random smoothing literature cannot be directly applied.

**Extension to priors supported on an affine subspace.** Our analysis naturally extends to the case where the prior distribution $\mu$ is supported on an affine subspace $S \subset \mathbb{R}^d$ of dimension $r \ll d$, representing a first step toward modelling the assumption that clean images lie on a low-dimensional manifold within the ambient space. Indeed, assuming that the restriction of $\mu$ to $S$ admits a positive log-concave density $p$ with respect to the $r$-dimensional Lebesgue measure on $S$, the smoothed density $p_\sigma$ is then defined over $\mathbb{R}^d$ and can naturally be decomposed into a Gaussian term orthogonal to $S$ and a convolution restricted to $S$. Specifically, for any point $z \in \mathbb{R}^d$, the smoothed density $p_\sigma(z)$ factorizes into a Gaussian penalty for the distance of $z$ to $S$, and the intrinsic smoothing of $p$ along $S$. Importantly, this decomposition allows us to express the third-order derivatives of $\ln p_\sigma$ in terms of derivatives intrinsic to $S$. As a result, Theorem 2 still holds but with ambient dimension $d$ replaced by the effective dimension $r \ll d$. We formally prove this in Section A.4.

**Extension when using approximate scores.** In practice we do not have access to the exact $\nabla \ln p_\sigma$ but only to an approximation of the score, often provided by a trained neural network $g_\sigma \approx \nabla \ln p_\sigma$. In this more realistic case, the MMSE Averaging recursion becomes $x_{k+1} = x_k - \alpha_k(\nabla F_{\sigma_k}(x_k) + \tau \xi_k)$ where $\xi_k := \nabla \ln p_{\sigma_k}(x_k) - g_{\sigma_k}(x_k)$ denotes the approximation error at step $k$. Assuming that these errors are uniformly bounded along the trajectory, i.e. $\|\xi_k\| \leq \xi$, we can show that the iterates converge to a point at distance $O(\xi)$ from the true proximal point, with the same rate as in Theorem 1. We refer to Section A.3 for the proof.

## 4 From Approximate Proximal Operators to MAP Estimation

We now return to the original MAP optimisation problem, recalled here:

$$\arg\min_{x \in \mathbb{R}^d} \lambda f(x) - \ln p(x).$$

We denote the objective by $J(x) := \lambda f(x) - \ln p(x)$ and work under the following assumption on the data fidelity term $f$:

**Assumption 3.** *The data fidelity term $f$ is convex, lower-bounded, and $L_f$-smooth.*

This is a mild assumption that holds for many common data fidelity terms, such as $f(x) = \frac{1}{2}\|Ax - y\|^2$ which is $L_f$-smooth with $L_f = 1/\lambda_{\max}(A^\top A)$. Note that we do not require $f$ to be strongly convex. Under this assumption, we denote $x_{\mathrm{MAP}}^\star \in \arg\min J$ any minimiser of $J$.

**Algorithm.** When the proximal operator is accessible, minimising $J$ can be achieved using proximal gradient descent, starting from $x^{(0)} = y$:

$$x^{(n+1)} = \mathrm{prox}_{-\tau \ln p}(x^{(n)} - \tau \lambda \nabla f(x^{(n)})). \tag{PGD}$$

Under Assumptions 1 and 3, the classical result of Beck and Teboulle [2009] (see their Theorem 3.1) guarantees that for a step size $\tau \leq 1/(\lambda L_f)$, the following convergence rate holds:

$$J(x^{(n)}) - J(x_{\mathrm{MAP}}^\star) \leq \frac{\|y - x_{\mathrm{MAP}}^\star\|^2}{2\tau n}.$$

In our setting, however, we do not have direct access to the exact proximal operator $\mathrm{prox}_{-\tau \ln p}$. Instead, we compute an approximate version using the MMSE Averaging recursion. Given a sequence $(k_n)_{n \geq 1}$ specifying the number of internal iterations used to approximate each proximal step, this leads naturally to an *inexact* proximal gradient descent algorithm.

---

**Algorithm 1** Approximate Proximal Gradient Descent (Approx PGD)

---

**Require:** Noisy image $y$, step size $\tau > 0$, parameter $\lambda > 0$, number of inner steps $(k_n)_{n \geq 1}$

    Initialise: $\hat{x}^{(0)} \leftarrow y$

    **for** $n = 0, 1, 2, \ldots$ **do**

        **1. Data fidelity gradient descent step**

        $z_0^{(n+1)} \leftarrow \hat{x}^{(n)} - \tau \lambda \nabla f(\hat{x}^{(n)})$

        **2. Approximate proximal step** $\hat{x}^{(n+1)} \approx \mathrm{prox}_{-\tau \ln p}(z_0^{(n+1)})$

        **for** $k = 0, \ldots, k_{n+1} - 1$ **do**

            $\sigma_k \leftarrow \sqrt{\frac{\tau}{k+1}}$

            $\alpha_k \leftarrow \frac{1}{k+2}$

            $z_{k+1}^{(n+1)} \leftarrow (1 - \alpha_k)\mathrm{MMSE}_{\sigma_k}(z_k^{(n+1)}) + \alpha_k z_0^{(n+1)}$

        **end for**

        $\hat{x}^{(n+1)} \leftarrow z_{k_{n+1}}^{(n+1)}$

    **end for**

---

We prove the following convergence result for the approximate proximal gradient descent iterates from Algorithm 1.

**Theorem 2** (Convergence towards the MAP estimator with explicit bounds). *For $\tau \leq \frac{1}{\lambda L_f}$ and a number of steps in the inner loop which increases as $k_n = \lfloor c \cdot n^{1+\eta} \rfloor$ for $c, \eta > 0$, under Assumptions 1 to 3 the approximate proximal gradient descent iterates $(\hat{x}^{(n)})_n$ from Algorithm 1 satisfy:*

$$\frac{1}{n} \sum_{i=1}^{n} J(x^{(i)}) - J(x_{\mathrm{MAP}}^{\star}) \leq O\left(\frac{1}{n}\right) \quad and \quad \|\hat{x}^{(n)} - x^{(n)}\| \leq \tilde{O}\left(\frac{1}{n^{1+\eta}}\right),$$

*where $x^{(n)} := \mathrm{prox}_{-\tau \ln p}(\hat{x}^{(n-1)} - \tau \lambda \nabla f(\hat{x}^{(n-1)}))$ denotes the exact proximal update at iteration $n$. The constants hidden in the $O(1/n)$ and $\tilde{O}(1/n)$ terms depend explicitly on the problem parameters and are given in detail in Section A.5.*

**Comment on the convergence bound.** This result provides a meaningful convergence guarantee in the context of MAP estimation. Since we do not assume strong convexity of $f$, it is more natural to measure progress through convergence in function value rather than in the iterates themselves. However, a direct bound on $J(\hat{x}^{(n)}) - J^{\star}$ cannot be expected in general: because the iterates $\hat{x}^{(n)}$ are only approximate updates of the true proximal points $x^{(n)}$, even a small error between $\hat{x}^{(n)}$ and $x^{(n)}$ can result in a large discrepancy in objective value due to the potentially poor conditioning of $J$. Instead, our analysis shows that the iterates $\hat{x}^{(n)}$ are close to the exact proximal iterates $x^{(n)}$, whose average MAP error is provably small. As a result, even though we cannot directly control $J(\hat{x}^{(n)})$, we ensure that the iterates are close to the iterates $x^{(n)}$ which provably converge (in average) towards the optimum.

**Sketch of proof.** We start from the classical descent inequality for proximal gradient updates. Since we use approximate proximal steps $\hat{x}^{(n)}$, we quantify the error $\varepsilon^{(n)} = \hat{x}^{(n)} - x^{(n)}$ using Theorem 1 and bound its impact on the objective. Summing over iterations and controlling the errors yields the $O(1/n)$ rate for the objective. The second bound follows directly from the convergence of the inner loop to the true proximal operator thanks to Theorem 1. Note that although our proof follows a similar strategy to that of Schmidt et al. [2011], which analyses inexact proximal gradient methods, their results do not directly apply here—because our approximation guarantee from Theorem 1 concern the iterates and not the objective function values.

Finally, note that while we consider an approximate version of proximal gradient descent, one could also analyse its accelerated counterpart, in the spirit of FISTA Beck and Teboulle [2009], which would yield faster convergence rates under the same assumptions. We leave this direction for future work.

# 5 Numerical Visualisations

To better understand the effect of smoothing on the proximal objective—and how it influences the gradient descent trajectory—we consider a simple two-dimensional example where the prior $p$ is a Gaussian distribution with a highly anisotropic covariance $\Sigma = \begin{pmatrix} 1 & 0 \\ 0 & 1/L \end{pmatrix}$ for $L \gg 1$. In this setting, the density $p(x_1, x_2)$ is sharply concentrated around the $x_1$-axis and rapidly decays as soon as $x_2 \neq 0$. The corresponding proximal objective $F(x) = \frac{1}{2}\|y - x\|^2 - \tau \ln p(x)$ is then a quadratic function with Hessian equal to

$$\nabla^2 F(x) = I + \tau \Sigma^{-1} = \begin{pmatrix} 1 + \tau & 0 \\ 0 & 1 + \tau L \end{pmatrix}.$$

As illustrated in Figure 2 this severe ill-conditioning leads gradient descent on $F$ to stagnate, making very little progress toward the true proximal point $\mathrm{prox}_{-\tau \ln p}(y)$.

However, smoothing the prior leads to a significant change in behaviour. Since $p_\sigma$ corresponds to the convolution of $p$ with a Gaussian of variance $\sigma^2$, it remains Gaussian with covariance $\Sigma_\sigma = \Sigma + \sigma^2 I_2$. The smoothed proximal objective $F_\sigma(x) = \frac{1}{2}\|y - x\|^2 - \tau \ln p_\sigma(x)$ is then also quadratic, but now with Hessian

$$\nabla^2 F_\sigma(x) = I + \tau \Sigma_\sigma^{-1} = \begin{pmatrix} 1 + \tau/(1 + \sigma^2) & 0 \\ 0 & 1 + \tau L/(1 + L\sigma^2) \end{pmatrix}.$$

As $\sigma$ increases, this Hessian interpolates between the poorly conditioned $\nabla^2 F$ and the well-conditioned identity matrix $I_2$. This transition is clearly visualised in Figure 1, which shows how the level curves of $F_\sigma$ become more isotropic as $\sigma$ increases. However, while smoothing improves conditioning, it also causes the minimiser $\mathrm{prox}_{-\tau \ln p_\sigma}(y) = \arg \min F_\sigma$ to drift away from the solution $\mathrm{prox}_{-\tau \ln p}(y) = \arg \min F$ which we ultimately aim to recover (the red triangle in Figure 1). This highlights the need for a decreasing schedule of $\sigma_k$ within the recursion: to benefit from better conditioning at early stages while still converging to the correct solution. This strategy leads to significantly improved optimisation performance. As shown in Figure 2, gradient descent applied to the sequence of smoothed objectives $(F_{\sigma_k})_k$, using the step size and noise schedule specified in Proposition 1, converges rapidly to the desired solution.

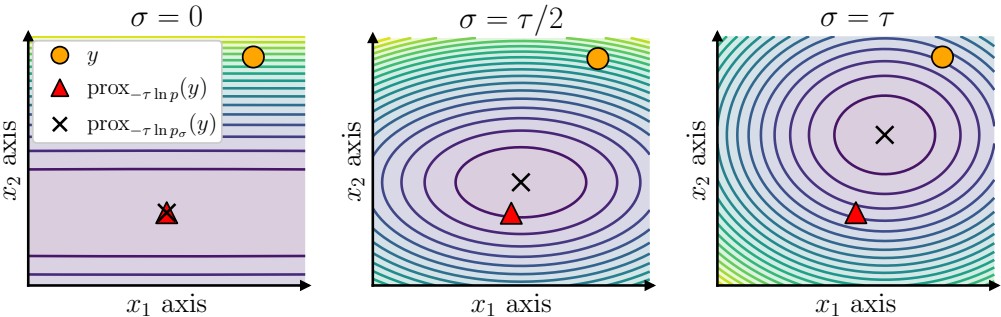

Figure 1: Visualisation of the level curves of the smoothed proximal objective $F_\sigma(x) = \frac{1}{2}\|y - x\|^2 - \tau \ln p_\sigma(x)$ for different values of $\sigma$. The unsmoothed objective $F$ is poorly conditioned *(left plot)*, but the conditioning improves significantly as $\sigma$ increases.

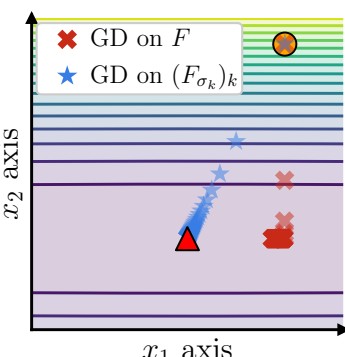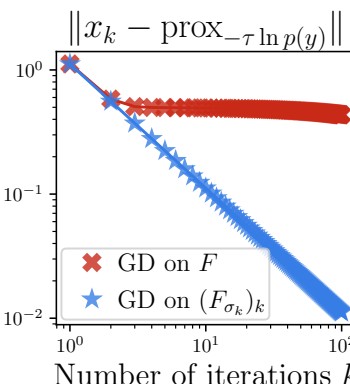

Figure 2: Illustration of the iterate trajectories *(left plot)* and convergence rates *(right plot)* of naive gradient descent on $F$ (which has condition number $\kappa = 500$) versus gradient descent on the smoothed objectives $(F_{\sigma_k})_k$, using a toy 2D Gaussian prior. Gradient descent on $F$, using a stepsize $\alpha = 0.8/L_F$ (chosen for better visualisation), suffers from poor conditioning and makes little progress toward the optimal solution $\text{prox}_{-\tau \ln p}(y)$. In contrast, gradient descent on the smoothed objectives $(F_{\sigma_k})_k$ converges rapidly, clearly exhibiting a $O(1/k)$ rate.

## 6  Conclusion

In this work, we prove that the iterative denoising-based scheme MMSE Averaging converges to the proximal operator of the negative log-prior $-\ln p$, a central component in MAP estimation for inverse problems. We show that, under suitable choices of averaging weights $\alpha_k$ and noise levels $\sigma_k$, the algorithm can be interpreted as gradient descent on a sequence of smoothed proximal objectives. Leveraging this perspective, we prove that the iterates converge to the true proximal point at a rate of $\tilde{O}(1/k)$, under the assumption that the prior $p$ is log-concave and has bounded third derivatives.

This result offers a principled foundation for a class of denoising-based schemes and connects them to classical optimisation theory. Importantly, it provides an explicit way to approximate the proximal operator of $-\ln p$, enabling the use of standard proximal methods to solve the MAP problem. We demonstrate this by incorporating our approximation into proximal gradient descent and deriving convergence guarantees for the resulting algorithm.

Despite these advances, our theoretical guarantees rely on strong assumptions — most notably that the prior is log-concave, sufficiently smooth, and supported on all of $\mathbb{R}^d$. Extending the analysis to more realistic settings, such as non-convex priors or those supported on low-dimensional manifolds, is an exciting direction for future work.

**Acknowledgements**

S. Pesme would like to warmly thank Filippo Santambrogio, who kindly and spontaneously responded to an email asking for help with Lemma 5. The proof we present is entirely based on the email exchange we had. S. Pesme is also grateful to Nikita Simonov, who generously replied to a similar email with several insightful suggestions for approaching the same lemma. Finally, S. Pesme would like to thank Loucas Pillaud-Vivien for the many valuable discussions they had regarding the proofs of Lemmas 5 and 6.

This work was supported by ERC grant number 101087696 (APHELEIA project) as well as by the ANR project BONSAI (grant ANR-23-CE23-0012-01).

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

## A  Proofs of Proposition 2 and Theorems 1 and 2

### A.1  Preliminary results

We start by the following proposition establishing that $-\ln p_\sigma$ is convex and $\frac{1}{\sigma^2}$-smooth.

**Proposition 4.** *Fix $\sigma > 0$. Under Assumption 1, $x \mapsto -\ln p_\sigma(x)$ is convex with a Hessian satisfying:*

$$-\nabla^2 \ln p_\sigma(z) = \frac{1}{\sigma^2}\big[I_d - \frac{1}{\sigma^2}\mathrm{Var}(\varepsilon|X + \sigma\varepsilon = z)\big] \preceq \frac{1}{\sigma^2}I_d.$$

*Proof.* The convexity of $x \mapsto -\ln p_\sigma(x)$ follows directly by the classical fact that a convolution of log-concave densities with a Gaussian is still log-concave (see [Saumard and Wellner, 2014, Proposition 3.5]). The fact that the Hessian is upper-bounded by $\frac{1}{\sigma^2}I_d$ is a direct consequence of an identity which can be seen as a "second order Tweedie formula" (e.g. Lemma A.1 in Gribonval [2011] or in Lee and Vázquez [2003] equation 5.8.):

$$-\nabla^2 \ln p_\sigma(z) = \frac{1}{\sigma^2}\big[I_d - \frac{1}{\sigma^2}\mathrm{Var}(\varepsilon|X + \sigma\varepsilon = z)\big]$$
$$\preceq \frac{1}{\sigma^2}I_d,$$

where $\varepsilon$ denotes a standard $d$-dimensional Gaussian random variable ($\varepsilon \sim \mathcal{N}(0, I_d)$) and the matrix inequality is due to the positiveness of the covariance matrix. For completeness we give the proof of the second order Tweedie identity. From the standard Tweedie identity (see, e.g. Efron [2011]) we have that:

$$-\nabla \ln p_\sigma(z) = \frac{z - \mathbb{E}\big[X|X + \sigma\varepsilon = z\big]}{\sigma^2}$$
$$= \frac{1}{\sigma^2}\int_{\mathbb{R}^d}(z - x)p(x|z)\mathrm{d}x$$
$$= \frac{1}{\sigma^2}\int_{\mathbb{R}^d}(z - x)\frac{\phi_\sigma(\|z - x\|)p(x)}{\int_{\mathbb{R}^d}\phi_\sigma(\|z - x'\|)p(x')\mathrm{d}x'}\mathrm{d}x,$$

where $\phi_\sigma(z) = \exp(-\frac{z^2}{2\sigma^2})$. Notice that $\phi_\sigma'(z) = -\frac{z}{\sigma^2}\phi_\sigma(z)$. We can now compute the Hessian of $-\ln p_\sigma$, letting $X_\sigma = X + \sigma\varepsilon$:

$$-\nabla^2 \ln p_\sigma(z) = \frac{1}{\sigma^2}\Big(I_d - \frac{1}{\sigma^2}\int_{\mathbb{R}^d}(z - x)^{\otimes 2}p(x|z)\mathrm{d}x + \frac{1}{\sigma^2}\big[\int_{\mathbb{R}^d}(z - x)p(x|z)\mathrm{d}x\big]^{\otimes 2}\Big)$$
$$= \frac{1}{\sigma^2}\Big(I_d - \frac{1}{\sigma^2}\big(\mathbb{E}[(X_\sigma - X)^{\otimes 2}|X_\sigma = z] - \mathbb{E}[X_\sigma - X|X_\sigma = z]^{\otimes 2}]\big)\Big)$$
$$= \frac{1}{\sigma^2}\Big(I_d - \frac{1}{\sigma^2}\mathrm{Var}(\varepsilon|X_\sigma = z)\Big),$$

which concludes the proof. □

Now, we recall and prove Proposition 2, which is a direct consequence of Proposition 4.

**Proposition 2.** *For any $\sigma > 0$, the function $F_\sigma$ is $L_\sigma$-smooth, with*

$$L_\sigma = 1 + \frac{\tau}{\sigma^2}.$$

*Proof.* The result directly follows from Proposition 4 which implies that $-\ln p_\sigma$ is $1/\sigma^2$-smooth, so that $F_\sigma$ is $L_\sigma$-smooth with $L_\sigma = 1 + \frac{\tau}{\sigma^2}$. □

## A.2 Analysis of the MMSE Averaging iterates

We start by recalling our main result Theorem 1, which provides a convergence rate towards the proximal operator of the MMSE Averaging recursion.

**Theorem 1** (Convergence to the Proximal operator). *Under Assumptions 1 and 2, let* $\mathrm{prox}_{-\tau \ln p}(y)$ *denote the unique solution of the Proximal Objective problem. Then, the MMSE Averaging iterates with parameters* $\alpha_k = 1/(k+2)$, $\sigma_k^2 = \tau/(k+1)$ *and initialised at* $x_0 = y$ *satisfy:*

$$\|x_k - \mathrm{prox}_{-\tau \ln p}(y)\| \leq \frac{(\ln k) + 7}{k+1}\big[\|y - \mathrm{prox}_{-\tau \ln p}(y)\| + \tau^2 M \sqrt{d}\big].$$

*Proof.* From Proposition 3, we are guaranteed that $F_{\sigma_k}$ is strongly convex and smooth with

$$\mu_{\sigma_k} = 1, \quad L_{\sigma_k} = 1 + \frac{\tau}{\sigma_k^2} = k+2, \quad \kappa_{\sigma_k} = k+2.$$

To avoid heavy notations, we denote $x_{\sigma_k}^\star := \mathrm{prox}_{-\tau \ln p_{\sigma_k}}(y) = \arg\min F_{\sigma_k}$ as well as $x^\star := \mathrm{prox}_{-\tau \ln p}(y) = \arg\min F$, note that these quantities are well defined and unique by the strong convexity of $F_{\sigma_k}$ and $F$.

Recall that due to Proposition 1, one step of the MMSE Averaging recursion can be seen as one step of gradient descent on $F_{\sigma_k}$ with stepsize $\alpha_k = \frac{1}{k+2}$, which exactly corresponds to $\alpha_k = 1/L_{\sigma_k}$. Hence, at iteration $k$, a standard convex optimisation result (see Theorem 2.1.15 in Nesterov [2013]) guarantees the contraction:

$$\|x_{k+1} - x_{\sigma_k}^\star\| \leq \left(1 - 2\frac{\mu_{\sigma_k}}{\mu_{\sigma_k} + L_{\sigma_k}}\right)^{1/2} \|x_k - x_{\sigma_k}^\star\|$$

$$= \left(\frac{\kappa_{\sigma_k} - 1}{\kappa_{\sigma_k} + 1}\right)^{1/2} \|x_k - x_{\sigma_k}^\star\|$$

$$= \left(\frac{k+1}{k+3}\right)^{1/2} \|x_k - x_{\sigma_k}^\star\| \tag{1}$$

We now use the triangle inequality to write:

$$\|x_{k+1} - x_{\sigma_k}^\star\| \leq \left(\frac{k+1}{k+3}\right)^{1/2} \big(\|x_k - x_{\sigma_{k-1}}^\star\| + \|x_{\sigma_{k-1}}^\star - x_{\sigma_k}^\star\|\big). \tag{2}$$

And we clearly see that we need to be able to control the regularity of $\sigma \mapsto x_\sigma^\star$. This is done in Proposition 9, where we show that $x_\sigma^\star$ is Lip schitz in $\sigma^2$:

$$\|x_{\sigma_1}^\star - x_{\sigma_2}^\star\|_2 \leq C(\sigma_1^2 - \sigma_2^2),$$

for $\sigma_2 \leq \sigma_1 \leq \sqrt{\tau}$ and where $C := \frac{1}{\tau}\|x^\star - y\| + \tau M \sqrt{d}$. Since $\sigma_k \leq \sqrt{\tau}$, we can use this bound and insert it in inequality (2) to get:

$$\|x_{k+1} - x_{\sigma_k}^\star\| \leq \left(\frac{k+1}{k+3}\right)^{1/2} \big(\|x_k - x_{\sigma_{k-1}}^\star\| + (\sigma_{k-1}^2 - \sigma_k^2) \cdot C\big).$$

It remains to unroll the inequality until $k=1$, and using the fact that $\Pi_{i=j}^k \left(\frac{i+1}{i+3}\right) = \frac{(j+1)(j+2)}{(k+2)(k+3)}$:

$$\|x_{k+1} - x_{\sigma_k}^\star\| \leq \frac{\sqrt{6}}{\sqrt{(k+2)(k+3)}}\|x_1 - x_{\sigma_0}^\star\| + \sum_{j=1}^k \sqrt{\frac{(j+1)(j+2)}{(k+2)(k+3)}}(\sigma_{j-1}^2 - \sigma_j^2)C.$$

And from inequality (1) we have that $\|x_1 - x_{\sigma_0}^\star\| \leq \frac{1}{\sqrt{3}}\|x_0 - x_{\sigma_0}^\star\|$. Since $x_0 = y$, this leads to:

$$\|x_{k+1} - x_{\sigma_k}^\star\| \leq \frac{\sqrt{2}}{\sqrt{(k+2)(k+3)}}\|y - x_{\sigma_0}^\star\| + \sum_{j=1}^k \sqrt{\frac{(j+1)(j+2)}{(k+2)(k+3)}}(\sigma_{j-1}^2 - \sigma_j^2)C.$$

Now since $\sigma_k^2 = \frac{\tau}{k+1}$, we have that $(\sigma_{j-1}^2 - \sigma_j^2) = \frac{\tau}{j(j+1)}$, hence for $k \geq 1$:

$$\|x_{k+1} - x_{\sigma_k}^\star\| \leq \frac{\sqrt{2}}{\sqrt{(k+2)(k+3)}}\|y - x_{\sigma_0}^\star\| + \sum_{j=1}^k \sqrt{\frac{(j+1)(j+2)}{(k+2)(k+3)}}\frac{\tau C}{j(j+1)}$$

$$\leq \frac{\sqrt{2}}{k+2}\|y - x_{\sigma_0}^\star\| + \frac{\tau C}{k+2}\sum_{j=1}^k \frac{j+2}{j(j+1)}$$

And we can simply bound:

$$\sum_{j=1}^k \frac{j+2}{j(j+1)} = \sum_{j=1}^k \left(\frac{1}{j} + \frac{1}{j} - \frac{1}{j+1}\right) \leq 1 + \sum_{j=1}^k \frac{1}{j} \leq 2 + \ln(k),$$

Therefore

$$\|x_{k+1} - x_{\sigma_k}^\star\| \leq \frac{\sqrt{2}}{k+2}\|y - x_{\sigma_0}^\star\| + \frac{(2 + \ln(k))\tau C}{k+2}.$$

Now using the triangular inequality $\|x_{k+1} - x^\star\| \leq \|x_{k+1} - x_{\sigma_k}^\star\| + \|x_{\sigma_k}^\star - x^\star\|$ and using Proposition 9 which bounds $\|x_{\sigma_k}^\star - x^\star\| \leq \sigma_k^2 C$ we get that:

$$\|x_{k+1} - x^\star\| \leq \frac{\sqrt{2}}{k+2}\|y - x_{\sigma_0}^\star\| + \frac{(2 + \ln(k))\tau C}{k+2} + \frac{\tau C}{k+1}.$$

And using the triangular inequality again:

$$\|y - x_{\sigma_0}^\star\| \leq \|y - x^\star\| + \|x^\star - x_{\sigma_0}^\star\|$$
$$\leq \|y - x^\star\| + \sigma_0^2 C$$
$$= \|y - x^\star\| + \tau C,$$

where the second inequality is due to Proposition 9. Therefore:

$$\|x_{k+1} - x^\star\| \leq \frac{\sqrt{2}\|y - x^\star\|}{k+2} + \frac{(\ln k) + 2 + \sqrt{2}}{k+1}\tau C,$$
$$\leq \frac{\sqrt{2}\|y - x^\star\|}{k+1} + \frac{(\ln k) + 4}{k+1}\tau C.$$

Plugging the definition of $C = \frac{1}{\tau}\|x^\star - y\| + \tau M\sqrt{d}$ we can finally write:

$$\|x_{k+1} - x^\star\| \leq \frac{(\ln k) + 7}{k+1}\left(\|x^\star - y\| + \tau^2 M\sqrt{d}\right),$$

which concludes the proof. $\qquad\square$

This next proposition proves the tightness of Theorem 1 (up to constants and the log-term) in the case of Gaussian prior. Here we assume that $p$ is the density of a $d$-dimensional Gaussian $\mathcal{N}(\mu, \Sigma)$, with $\Sigma$ a positive definite matrix. Without loss of generality, we can assume that the Gaussian is centered: i.e., $\mu = 0$.

**Proposition 5** (Exact convergence rate for Gaussian priors.)**.** *Under the assumption that the prior $p$ is a $d$-dimensional centered Gaussian $\mathcal{N}(0, \Sigma)$, then we have that the MMSE Averaging recursion with $\alpha_k = 1/(k+2)$, $\sigma_k^2 = \tau/(k+1)$ initialised at $x_0 = y$ satisfies the identity:*

$$x_k - \mathrm{prox}_{-\tau \ln p}(y) = \frac{y - \mathrm{prox}_{-\tau \ln p}(y)}{k+1}.$$

*Proof.* In this setting, the negative log-prior $-\ln p$ is a quadratic with Hessian $H = \Sigma^{-1}$, and $F$ is a quadratic:

$$F(x) = \frac{1}{2}\|y - x\|^2 + \frac{\tau}{2}x^\top \Sigma^{-1} x.$$

Its minimiser is given by:

$$x^\star := \mathrm{prox}_{-\tau \ln p}(y) = (I + \tau \Sigma^{-1})^{-1} y.$$

And since $p_\sigma \sim \mathcal{N}(0, \Sigma + \sigma^2 I_d)$, the smoothed objective writes:

$$F_{\sigma_k}(x) = \frac{1}{2}\|y - x\|^2 + \frac{\tau}{2} x^\top (\Sigma + \sigma_k^2 I_d)^{-1} x,$$

and its gradient is:

$$\nabla F_{\sigma_k}(x) = x - y + \tau (\Sigma + \sigma_k^2 I_d)^{-1} x.$$

We now prove the result by induction. For $k = 0$, we have $x_0 = y$ and the base case trivially holds.

**Inductive step:** The inductive hypothesis provides that:

$$x_k = x^\star + \frac{1}{k+1}(y - x^\star).$$

Using the identity $x^\star = (I + \tau \Sigma^{-1})^{-1} y$, we have:

$$y - x^\star = \tau \Sigma^{-1} x^\star \quad \Rightarrow \quad x_k = x^\star + \frac{\tau}{k+1}\Sigma^{-1} x^\star.$$

Then,

$$(\Sigma + \sigma_k^2 I_d)^{-1} x_k = (\Sigma + \tfrac{\tau}{k+1} I_d)^{-1}\left(I + \tfrac{\tau}{k+1}\Sigma^{-1}\right) x^\star = \Sigma^{-1} x^\star = \frac{y - x^\star}{\tau},$$

so that:

$$\nabla F_{\sigma_k}(x_k) = x_k - y + (y - x^\star) = x^\star - y + \frac{1}{k+1}(y - x^\star) + (y - x^\star) = \frac{y - x^\star}{k+1}$$

Now from Proposition 1, the update writes:

$$
\begin{aligned}
x_{k+1} &= x_k - \frac{1}{k+2}\nabla F_{\sigma_k}(x_k) \\
&= x^\star + \frac{y - x^\star}{k+1} - \frac{y - x^\star}{(k+1)(k+2)} \\
&= x^\star + \frac{(y - x^\star)}{k+2}.
\end{aligned}
$$

This completes the inductive step, and hence the proof. $\qquad \square$

### A.3   Extension when using approximate scores

In practice, when using a trained denoiser, we do not have access to the exact score $\nabla \ln p_\sigma$, but only to an approximation $g_\sigma \approx \nabla \ln p_\sigma$. In this more realistic case, the update rule becomes:

$$x_{k+1} = x_k - \alpha_k\big(x_k - y - \tau g_{\sigma_k}(x_k)\big)$$

where we use the approximation $g_\sigma$ instead of the true score $-\nabla \ln p_\sigma$. This recursion rewrites

$$x_{k+1} = x_k - \alpha_k(\nabla F_{\sigma_k}(x_k) + \tau \xi_k) \qquad\qquad \text{(Noisy recursion)}$$

where $\xi_k := \nabla \ln p_{\sigma_k}(x_k) - g_{\sigma_k}(x_k)$ denotes the approximation error at step $k$. Assuming that these errors are uniformly bounded along the trajectory, i.e. $\|\xi_k\| \leq \xi$, we can show that the iterates converge to a point at distance $O(\xi)$ from the true proximal point, with the same rate as in Theorem 1.

**Proposition 6** (Convergence with approximate scores). *Under Assumptions 1 and 2, let* $\mathrm{prox}_{-\tau \ln p}(y)$ *denote the unique solution of the Proximal Objective problem. If the score approximation errors satisfy* $\|\xi_k\| \leq \xi$ *for all* $k$, *then the Noisy recursion iterates with parameters* $\alpha_k = 1/(k+2)$, $\sigma_k^2 = \tau/(k+1)$ *and initialised at* $x_0 = y$ *satisfy:*

$$\|x_k - \mathrm{prox}_{-\tau \ln p}(y)\| \leq \frac{(\ln k) + 7}{k+1}\big[\|y - \mathrm{prox}_{-\tau \ln p}(y)\| + \tau^2 M\sqrt{d}\big] + \sqrt{\tfrac{3}{2}}\tau \xi.$$

*Proof.* To avoid heavy notations, we denote $x^\star_{\sigma_k} := \operatorname{prox}_{-\tau \ln p_{\sigma_k}}(y) = \arg\min F_{\sigma_k}$ as well as $x^\star := \operatorname{prox}_{-\tau \ln p}(y) = \arg\min F$, note that these quantities are well defined and unique by the strong convexity of $F_{\sigma_k}$ and $F$.

Let $\tilde{x}_{k+1} := x_k - \alpha_k \nabla F_{\sigma_k}(x_k)$ be the noiseless step. Following the exact same arguments as in the proof of Theorem 1, we get the single–step contraction

$$\|\tilde{x}_{k+1} - x^\star_{\sigma_k}\| \le \Big(\frac{k+1}{k+3}\Big)^{1/2} \|x_k - x^\star_{\sigma_k}\|.$$

Since $x_{k+1} = \tilde{x}_{k+1} - \alpha_k \tau \xi_k$, the triangle inequality leads to:

$$\|x_{k+1} - x^\star_{\sigma_k}\| \le \Big(\frac{k+1}{k+3}\Big)^{1/2} \|x_k - x^\star_{\sigma_k}\| + \frac{\tau}{k+2} \|\xi_k\|. \tag{3}$$

Next, as in the noiseless case, we decompose as:

$$\|x_k - x^\star_{\sigma_k}\| \le \|x_k - x^\star_{\sigma_{k-1}}\| + \|x^\star_{\sigma_{k-1}} - x^\star_{\sigma_k}\| \le \|x_k - x^\star_{\sigma_{k-1}}\| + C(\sigma^2_{k-1} - \sigma^2_k),$$

with $C = \frac{1}{\tau}\|x^\star - y\| + \tau M \sqrt{d}$ and $\sigma^2_{k-1} - \sigma^2_k = \frac{\tau}{k(k+1)}$. Plugging this into the previous inequality and unrolling from $j = 1$ to $k$ gives

$$\|x_{k+1} - x^\star_{\sigma_k}\| \le \frac{\sqrt{6}}{\sqrt{(k+2)(k+3)}} \|x_1 - x^\star_{\sigma_0}\| + \sum_{j=1}^k \sqrt{\frac{(j+1)(j+2)}{(k+2)(k+3)}} (\sigma^2_{j-1} - \sigma^2_j) C$$
$$+ \sum_{j=1}^k \sqrt{\frac{(j+2)(j+3)}{(k+2)(k+3)}} \frac{\tau}{j+2} \|\xi_j\|.$$

From inequality (3) with $k = 0$, we have that $\|x_1 - x^\star_{\sigma_0}\| \le \frac{1}{\sqrt{3}}\|x_0 - x^\star_{\sigma_0}\| + \frac{\tau}{2}\|\xi_0\|$. Since $x_0 = y$, we get:

$$\|x_{k+1} - x^\star_{\sigma_k}\| \le \frac{\sqrt{2}}{\sqrt{(k+2)(k+3)}} \|y - x^\star_{\sigma_0}\| + \sum_{j=1}^k \sqrt{\frac{(j+1)(j+2)}{(k+2)(k+3)}} (\sigma^2_{j-1} - \sigma^2_j) C$$
$$+ \sum_{j=0}^k \sqrt{\frac{(j+2)(j+3)}{(k+2)(k+3)}} \frac{\tau}{j+2} \|\xi_j\|.$$

The second sum is bounded exactly as in the noiseless case:

$$\sum_{j=1}^k \sqrt{\frac{(j+1)(j+2)}{(k+2)(k+3)}} \frac{\tau C}{j(j+1)} \le \frac{(2 + \ln k)\tau C}{k+2}.$$

For the noise sum, using $\|\xi_j\| \le \xi$ and $\sqrt{\frac{(j+2)(j+3)}{(k+2)(k+3)}} \frac{1}{j+2} \le \frac{\sqrt{3/2}}{\sqrt{(k+2)(k+3)}}$, we obtain

$$\sum_{j=0}^k \sqrt{\frac{(j+2)(j+3)}{(k+2)(k+3)}} \frac{\tau}{j+2} \|\xi_j\| \le \frac{\sqrt{3/2}(k+1)\tau\xi}{\sqrt{(k+2)(k+3)}} \le \sqrt{\tfrac{3}{2}}\tau\xi.$$

Putting things together, exactly as in the proof of Theorem 1, we obtain for all $k \ge 1$:

$$\|x_{k+1} - x^\star\| \le \frac{(\ln k) + 7}{k+1} \Big(\|x^\star - y\| + \tau^2 M \sqrt{d}\Big) + \sqrt{\tfrac{3}{2}}\tau\xi.$$

Thus, the iterates converge to an $O(\xi)$ neighbourhood of $\operatorname{prox}_{-\tau \ln p}(y)$ with the same $O(1/k)$ rate as in the noiseless case. $\qquad\square$

## A.4 Extension to distributions supported on affine subspaces of $\mathbb{R}^d$

In this subsection we prove that Theorem 1 can naturally be extended to the case where the prior distribution is supported on an affine subspace of dimension $r \ll d$, in which case the dimension $d$ which appears in the upperbound reduces to the effective dimension $r$. Formally, we assume that the clean images $x$ are drawn from a probability distribution $\mu$ on $\mathbb{R}^d$ satisfying the following:

**Assumption 4.** *There exists an affine subspace $S \subset \mathbb{R}^d$ of dimension $r \leq d$ such that the probability distribution $\mu \in \mathcal{P}(\mathbb{R}^d)$ satisfies:*

- *$\mu$ is supported on $S$: $\mu(\mathbb{R}^d \setminus S) = 0$. Moreover, the restriction of $\mu$ to $S$ admits a density $p : S \to \mathbb{R}_+$ with respect to the $r$-dimensional Lebesgue measure on $S$. By abuse of notation, we extend $p$ to $\mathbb{R}^d$ by setting $p(x) = 0$ for $x \in \mathbb{R}^d \setminus S$.*

- *$p(x) > 0$ for all $x \in S$.*

- *$p$ is log-concave.*

Let $\phi_\sigma(x) = \exp\left(-\frac{\|x\|^2}{2\sigma^2}\right)$ denote the Gaussian kernel on $\mathbb{R}^d$ of variance $\sigma^2$, now let $C_\sigma := (2\pi\sigma^2)^{1/2}$ such that $\int_{\mathbb{R}^d} \phi_\sigma(x) = C_\sigma^d$. The smoothed density function $p_\sigma : \mathbb{R}^d \to \mathbb{R}_+$ then writes, for all $z \in \mathbb{R}^d$:

$$p_\sigma(z) = \frac{1}{C_\sigma^d} \int_{\mathbb{R}^d} \phi_\sigma(z - x)\,\mathrm{d}\mu(x)$$

$$= \frac{1}{C_\sigma^d} \int_S p(x)\,\phi_\sigma(z - x)\,\mathrm{d}x.$$

For $z \in \mathbb{R}^d$, let $z_\perp$ denote the orthogonal projection of $z$ on $S$. Using orthogonality, notice that:

$$p_\sigma(z) = \frac{\phi_\sigma(z - z_\perp)}{C_\sigma^{d-r}} \cdot \frac{1}{C_\sigma^r} \int_S p(x)\phi_\sigma(z_\perp - x)\mathrm{d}x.$$

Therefore, for $z \in S$, letting $\tilde{p}_\sigma(z) := \frac{1}{C_\sigma^p} \int_S p(x)\phi_\sigma(z - x)\mathrm{d}x$ denote the convolution of $p$ with the Gaussian kernel over $S$, we get that

$$-\ln p_\sigma(z) = \frac{\|z - z_\perp\|_2^2}{2\sigma^2} - \ln \tilde{p}_\sigma(z_\perp) + (d - r)\ln C_\sigma.$$

And importantly:

$$-\nabla\Delta\ln p_\sigma(z) = -\nabla_S\Delta_S\ln\tilde{p}_\sigma(z_\perp),$$

where the $\nabla_S$ and $\Delta_S$ denote the intrinsic gradients and Laplacians on $S$.

Therefore using Lemma 5 for $\tilde{p}_\sigma$ we have the following upper bound:

$$\sup_{z\in\mathbb{R}^d} \|\nabla\Delta\ln p_\sigma(z)\| = \sup_{z_\perp\in S} \|\nabla_S\Delta_S\ln\tilde{p}_\sigma(z_\perp)\|$$

$$\leq \sqrt{r} \sup_{z_\perp\in S} \|\nabla_S^3\ln p(z_\perp)\|.$$

From this point onward, the proof of Theorem 1 carries through, with the ambient dimension $d$ replaced by the effective dimension $r$.

## A.5 Analysis of the approximate PGD Algorithm 1

We now restate and prove the convergence of the approximate PGD algorithm towards the MAP estimator. The following is a restatement of Theorem 2 with explicit constants.

**Theorem 3** (Convergence towards the MAP estimator with explicit bounds). *For $\tau \leq \frac{1}{\lambda L_f}$ and a number of steps in the inner loop which increases as $k_n = \lfloor c \cdot n^{1+\eta} \rfloor$ for $c, \eta > 0$, the approximate proximal gradient descent iterates $(\hat{x}^{(n)})_n$ from Algorithm 1 satisfy:*

$$\frac{1}{n}\sum_{i=1}^n J(x^{(i)}) - J(x^\star_{\mathrm{MAP}}) \leq \frac{1}{2\tau n}\left(\|y - x^\star_{\mathrm{MAP}}\|^2 + \sum_{i=1}^\infty \|\varepsilon_i\|^2 + 2R_{\eta,c}\sum_{i=1}^\infty \|\varepsilon_i\|\right)$$

$$\|\hat{x}^{(n)} - x^{(n)}\| \leq \frac{(1+\eta)\ln(n) + \ln(c) + 7}{c \cdot n^{1+\eta}} \cdot R_{\eta,c},$$

where $x^{(n)} := \operatorname{prox}_{-\tau \ln p}(\hat{x}^{(n-1)} - \tau\lambda\nabla f(\hat{x}^{(n-1)}))$ *corresponds to the true proximal mapping, and where the quantities* $R_{\eta,c}$, $\sum_{i=1}^{\infty}\|\varepsilon_i\|$ *and* $\sum_{i=1}^{\infty}\|\varepsilon_i\|^2$ *are explicitly upper bounded in Lemma* 1.

*For, e.g.,* $\eta = 1$ *and* $c = 10$*, the bounds become:*

$$\frac{1}{n}\sum_{i=1}^{n} J(x^{(i)}) - J^{\star} \lesssim \frac{1}{\tau k}\Big(300\cdot\|y - x_{\mathrm{MAP}}^{\star}\|^2 + 600\cdot\big(\tau\lambda\|\nabla f(x_{\mathrm{MAP}}^{\star})\| + \tau^2 M\sqrt{d}\big)\Big)$$

$$\|\hat{x}^{(n)} - x^{(n)}\| \lesssim \frac{2\ln(n) + 10}{n^2}\cdot\Big(6\cdot\|y - x_{\mathrm{MAP}}^{\star}\|^2 + 12\cdot\big(\tau\lambda\|\nabla f(x_{\mathrm{MAP}}^{\star})\| + \tau^2 M\sqrt{d}\big)\Big).$$

*Proof.* For $\tau \le \frac{1}{\lambda L_f}$, the classic inequality after one step of the true proximal descent $x^{(n+1)} := \operatorname{prox}_{-\tau \ln p}(\hat{x}^{(n)} - \tau\lambda\nabla f(\hat{x}^{(n)}))$ provides that (see, e.g. equation 3.6 in Beck and Teboulle [2009]):

$$J(x^{(n+1)}) - J^{\star} \le \frac{1}{2\tau}(\|\hat{x}^{(n)} - x_{\mathrm{MAP}}^{\star}\|^2 - \|x^{(n+1)} - x_{\mathrm{MAP}}^{\star}\|^2). \tag{4}$$

Now for $n \ge 1$, let $\varepsilon_n := \hat{x}^{(n)} - x^{(n)}$ correspond to approximation error which can be quantified using Theorem 1. Letting $J^{\star} := J(x_{\mathrm{MAP}}^{\star})$, for $n \ge 1$, inequality (4) can be expanded as:

$$J(x^{(n+1)}) - J^{\star} \le \frac{1}{2\tau}\Big(\|x^{(n)} - x_{\mathrm{MAP}}^{\star}\|^2 - \|x^{(n+1)} - x_{\mathrm{MAP}}^{\star}\|^2 + \|\hat{x}^{(n)} - x^{(n)}\|^2 + 2\langle \hat{x}^{(n)} - x^{(n)}, x^{(n)} - x_{\mathrm{MAP}}^{\star}\rangle\Big)$$

$$\le \frac{1}{2\tau}\Big(\|x^{(n)} - x_{\mathrm{MAP}}^{\star}\|^2 - \|x^{(n+1)} - x_{\mathrm{MAP}}^{\star}\|^2 + \|\varepsilon_n\|^2 + 2\|\varepsilon_n\|\cdot\|x^{(n)} - x_{\mathrm{MAP}}^{\star}\|\Big)$$

$$\le \frac{1}{2\tau}\Big(\|x^{(n)} - x_{\mathrm{MAP}}^{\star}\|^2 - \|x^{(n+1)} - x_{\mathrm{MAP}}^{\star}\|^2 + \|\varepsilon_n\|^2 + 2R_{\eta,c}\|\varepsilon_n\|\Big),$$

where the second inequality is due to the Cauchy-Schwarz inequality, and the bound $\|x^{(n)} - x_{\mathrm{MAP}}^{\star}\| \le R_{\eta,c}$ is due to Lemma 1. It remains to sum this inequality from $i = 1$ to $n - 1$ and add inequality 4 with $n = 0$ to get:

$$\sum_{i=1}^{n}(J(x^{(i)}) - J^{\star}) \le \frac{1}{2\tau}\Big(\|\hat{x}_0 - x_{\mathrm{MAP}}^{\star}\|^2 - \|x^{(n)} - x_{\mathrm{MAP}}^{\star}\|^2 + \sum_{i=1}^{n-1}\|\varepsilon_i\|^2 + 2R_{\eta,c}\sum_{i=1}^{n-1}\|\varepsilon_i\|\Big)$$

$$\le \frac{1}{2\tau}\Big(\|y - x_{\mathrm{MAP}}^{\star}\|^2 + \sum_{i=1}^{\infty}\|\varepsilon_i\|^2 + 2R_{\eta,c}\sum_{i=1}^{\infty}\|\varepsilon_i\|\Big)$$

where the second inequality is due to Lemma 1. Diving by $n$ leads to the first result. The second comes from the fact that $\|\varepsilon_n\| = \|\hat{x}^{(n)} - x^{(n)}\|$ for which the upper bound is given in Lemma 1. $\square$

The following lemma provides a bound on this approximation error at each step, along with bounds on other useful quantities.

**Lemma 1.** *For* $\tau \le \frac{1}{\lambda L_f}$ *and a number of steps in the inner loop which increases as* $k_n = \lfloor c\cdot n^{1+\eta}\rfloor$ *for* $c, \eta > 0$*, let* $(\hat{x}^{(n)})_n$ *denote the approximate proximal gradient descent iterates from Algorithm* 1 *and let* $\varepsilon_n := \hat{x}^{(n)} - x^{(n)}$ *denote the approximation error at iteration* $n$*, where* $x^{(n)} := \operatorname{prox}_{-\tau \ln p}(\hat{x}^{(n-1)} - \tau\lambda\nabla f(\hat{x}^{(n-1)}))$ *is the true proximal point. Then it holds that:*

$$\|x^{(n)} - x_{\mathrm{MAP}}^{\star}\| \le R_{\eta,c}, \qquad \|\varepsilon_n\| \le \frac{(1+\eta)\ln(n) + \ln(c) + 7}{c\cdot n^{1+\eta}}\cdot R_{\eta,c},$$

$$\sum_{n=1}^{\infty}\|\varepsilon_n\| \le S_{\eta,c}\cdot R_{\eta,c}, \quad \sum_{n=1}^{\infty}\|\varepsilon_n\|^2 \le T_{\eta,c}\cdot R_{\eta,c}^2.$$

*where*

$$R_{\eta,c} := B_{\eta,c} + \tau\lambda\|\nabla f(x_{\mathrm{MAP}}^{\star})\| + \tau^2 M\sqrt{d}$$

$$B_{\eta,\sigma} := \exp(2S_{\eta,c})\Big[\|y - x_{\mathrm{MAP}}^{\star}\| + S_{\eta,c}\cdot\big(\tau\lambda\|\nabla f(x_{\mathrm{MAP}}^{\star})\| + \tau^2 M\sqrt{d}\big)\Big]$$

$$S_{\eta,c} := \frac{1+\eta}{c\eta^2}\big(1 + \eta\cdot(\ln(c) + 7)\big)$$

$$T_{\eta,c} := \frac{4(1+\eta)^2}{c^2(2\eta+1)^3} + \frac{2(\ln(c)+7)^2}{c^2}\Big(1 + \frac{1}{2\eta+1}\Big)$$

*For, e.g., $\eta = 1$, $c = 10$, these quantities simply become:*

$$R_{\eta,c} \approx B_{\eta,\sigma} \approx 60 \cdot \|y - x_{\mathrm{MAP}}^\star\| + 120 \cdot \left(\tau\lambda\|\nabla f(x_{\mathrm{MAP}}^\star)\| + \tau^2 M\sqrt{d}\right)$$
$$S_{\eta,c} \approx T_{\eta,c} \approx 2$$

*Proof.* From inequality (4), for $n \geq 1$ we have that:

$$\|x^{(n)} - x_{\mathrm{MAP}}^\star\| \leq \|\hat{x}^{(n-1)} - x_{\mathrm{MAP}}^\star\| \tag{5}$$
$$\leq \|\hat{x}^{(n-1)} - x^{(n-1)}\| + \|x^{(n-1)} - x_{\mathrm{MAP}}^\star\|$$
$$= \|\varepsilon_{n-1}\| + \|x^{(n-1)} - x_{\mathrm{MAP}}^\star\|.$$

Furthermore, from Theorem 1, since $c \cdot n^{1+\eta} - 1 \leq k_n \leq c \cdot n^{1+\eta}$, we get for $n \geq 1$:

$$\|\varepsilon_n\| := \|\hat{x}^{(n)} - x^{(n)}\| \leq \frac{(\ln k_n) + 7}{k_n + 1}\left[\|\hat{x}^{(n-1)} - \tau\lambda\nabla f(\hat{x}^{(n-1)}) - x^{(n)}\| + \tau^2 M\sqrt{d}\right]$$
$$\leq \frac{(1+\eta)\ln(n) + \ln(c) + 7}{c \cdot n^{1+\eta}}\left[\|x^{(n)} - (I_d - \tau\lambda\nabla f)(\hat{x}^{(n-1)})\| + \tau^2 M\sqrt{d}\right]. \tag{6}$$

Now, we use the triangle inequality to write:

$$\|x^{(n)} - (I_d - \tau\lambda\nabla f)(\hat{x}^{(n-1)})\|$$
$$\leq \|x^{(n)} - x_{\mathrm{MAP}}^\star\| + \|x_{\mathrm{MAP}}^\star - (I_d - \tau\lambda\nabla f)(x_{\mathrm{MAP}}^\star)\| \tag{7}$$
$$+ \|(I_d - \tau\lambda f)(x_{\mathrm{MAP}}^\star) - (I_d - \tau\lambda f)(\hat{x}^{(n-1)})\|$$

Now, since $x_{\mathrm{MAP}}^\star$ satisfies the fixed point property $x_{\mathrm{MAP}}^\star = \mathrm{prox}_{-\tau \ln p}((I_d - \tau\lambda\nabla f)(x_{\mathrm{MAP}}^\star))$, and from the definition of $x^{(n)}$, we can write:

$$\|x^{(n)} - x_{\mathrm{MAP}}^\star\| = \|\mathrm{prox}_{-\tau \ln p}\left((I_d - \tau\lambda\nabla f)(\hat{x}^{(n-1)})\right) - \mathrm{prox}_{-\tau \ln p}\left((I_d - \tau\lambda\nabla f)(x_{\mathrm{MAP}}^\star)\right)\|$$
$$\leq \|(I_d - \tau\lambda\nabla f)(\hat{x}^{(n-1)}) - (I_d - \tau\lambda\nabla f)(x_{\mathrm{MAP}}^\star)\|,$$

where the inequality is due to the non-expansiveness of the proximal operator. Inequality 7 then becomes

$$\|x^{(n)} - (I_d - \tau\lambda\nabla f)(\hat{x}^{(n-1)})\| \leq 2\|(I_d - \tau\lambda f)(x_{\mathrm{MAP}}^\star) - (I_d - \tau\lambda f)(\hat{x}^{(n-1)})\| + \tau\lambda\|\nabla f(x_{\mathrm{MAP}}^\star)\|$$
$$\leq 2\|x_{\mathrm{MAP}}^\star - \hat{x}^{(n-1)}\| + \tau\lambda\|\nabla f(x_{\mathrm{MAP}}^\star)\|,$$

where the second inequality is because $I_d - \tau\lambda f$ is Lipschitz for $\tau \leq 1/(\lambda L_f)$. Therefore, injecting this bound in the inequality 6, we get for $n \geq 1$:

$$\|\varepsilon_n\| \leq \frac{(1+\eta)\ln(n) + \ln(c) + 7}{c \cdot n^{1+\eta}}\left[2\|\hat{x}^{(n-1)} - x_{\mathrm{MAP}}^\star\| + \tau\lambda\|\nabla f(x_{\mathrm{MAP}}^\star)\| + \tau^2 M\sqrt{d}\right]$$
$$\leq \frac{(1+\eta)\ln(n) + \ln(c) + 7}{c \cdot n^{1+\eta}}\left[2\|\varepsilon_{n-1}\| + 2\|x^{(n-1)} - x_{\mathrm{MAP}}^\star\| + \tau\lambda\|\nabla f(x_{\mathrm{MAP}}^\star)\| + \tau^2 M\sqrt{d}\right]. \tag{8}$$

where the second inequality still holds for $n = 1$ with the convention $\varepsilon_0 = 0$ and $x_0 = \hat{x}_0 = y$. Now adding the inequality $\|x^{(n)} - x_{\mathrm{MAP}}^\star\| \leq \|\varepsilon_{n-1}\| + \|x^{(n-1)} - x_{\mathrm{MAP}}^\star\|$ from inequality (5) to the above inequality 8, and letting $w_n := \|\varepsilon_n\| + \|x^{(n)} - x_{\mathrm{MAP}}^\star\|$ for $n \geq 0$, we get the following recursive inequality for $n \geq 1$:

$$w_n \leq (1 + 2C_n)w_{n-1} + C_n A,$$

where

$$C_n := \frac{(1+\eta)\ln(n) + \ln(c) + 7}{c \cdot n^{1+\eta}}, \quad A := \tau\lambda\|\nabla f(x_{\mathrm{MAP}}^\star)\| + \tau^2 M\sqrt{d}, \quad w_0 = \|y - x_{\mathrm{MAP}}^\star\|.$$

It now remains to unroll the recursive inequality on $w_n$, which is done in the auxiliary Lemma 2 to obtain:

$$w_n \leq \exp(2S_{\eta,c})\left(w_0 + AS_{\eta,c}\right),$$

where

$$S_{\eta,c} := \frac{1+\eta}{c\eta^2}\big(1 + \eta \cdot (\ln(c) + 7)\big),$$

Putting things together we get the following uniform bound on $w_n$:

$$w_n \leq B_{\eta,\sigma} := \exp(2S_{\eta,c})\Big[\|y - x_{\text{MAP}}^\star\| + S_{\eta,c} \cdot \big(\tau\lambda\|\nabla f(x_{\text{MAP}}^\star)\| + \tau^2 M\sqrt{d}\big)\Big]$$

From the definition of $w_n = \|\varepsilon_n\| + \|x^{(n)} - x_{\text{MAP}}^\star\|$, we trivially get that $\|x^{(n)} - x_{\text{MAP}}^\star\| \leq B_{\eta,c}$, and now from inequality (8) we get, for $n \geq 1$:

$$\|\varepsilon_n\| \leq \frac{(1+\eta)\ln(n) + \ln(c) + 7}{c \cdot n^{1+\eta}}\big[2B_{\eta,c} + \tau\lambda\|\nabla f(x_{\text{MAP}}^\star)\| + \tau^2 M\sqrt{d}\big].$$

Letting $R_{\eta,c} := 2B_{\eta,c} + \tau\lambda\|\nabla f(x_{\text{MAP}}^\star)\| + \tau^2 M\sqrt{d} \geq B_{\eta,c}$ we prove the two first inequalities of the statement.

Now to bound $\sum_{n=1}^\infty \|\varepsilon_n\|$ we simply reuse the bound obtained on $\sum_i C_i \leq S_{\eta,c}$ in the proof of Lemma 2 to obtain:

$$\sum_{n=1}^\infty \|\varepsilon_n\| \leq S_{\eta,c} \cdot R_{\eta,c}.$$

Finally for $\sum_{n=1}^\infty \|\varepsilon_n\|^2$ we upperbound:

$$\sum_{n=1}^\infty \left(\frac{(1+\eta)\ln(n) + \ln(c) + 7}{c \cdot n^{1+\eta}}\right)^2 \leq \frac{2(1+\eta)^2}{c^2}\sum_{n=1}^\infty \frac{\ln^2(n)}{n^{2(1+\eta)}} + \frac{2(\ln(c)+7)^2}{c^2}\sum_{n=1}^\infty \frac{1}{n^{2(1+\eta)}}.$$

We now bound the two series using integrals:

$$\sum_{n=1}^\infty \frac{\ln^2(k)}{n^{2(1+\eta)}} \leq \int_1^\infty \frac{\ln^2(x)}{x^{2(1+\eta)}}\,dx = \frac{2}{(2\eta+1)^3},$$

$$\sum_{n=1}^\infty \frac{1}{n^{2(1+\eta)}} \leq 1 + \int_1^\infty \frac{1}{x^{2(1+\eta)}}\,dx = 1 + \frac{1}{2\eta+1}.$$

Putting everything together, we obtain the bound:

$$\sum_{n=1}^\infty \|\varepsilon_n\|^2 \leq \left(\frac{4(1+\eta)^2}{c^2(2\eta+1)^3} + \frac{2(\ln(c)+7)^2}{c^2}\left(1 + \frac{1}{2\eta+1}\right)\right)R_{\eta,c}^2,$$

which concludes the proof. $\qquad\square$

**Lemma 2.** *The recursive inequality*

$$w_n \leq (1 + 2C_n)w_{n-1} + C_n A, \quad where \quad C_n := \frac{(1+\eta)\ln(n) + \ln(c) + 7}{c \cdot n^{1+\eta}}$$

*unrolls as:*

$$w_n \leq \exp(2S_{\eta,c})\left(w_0 + AS_{\eta,c}\right),$$

*where*

$$S_{\eta,c} := \frac{1+\eta}{c\eta^2}\big(1 + \eta \cdot (\ln(c) + 7)\big).$$

*Proof.* We iteratively apply the inequality to obtain:

$$w_n \leq w_0 \prod_{j=1}^n (1 + 2C_j) + A\sum_{i=1}^n C_i \prod_{j=i+1}^n (1 + 2C_j),$$

with the convention that empty products are equal to 1.

We now bound the product $\prod_{j=1}^n (1 + 2C_j)$ by using the inequality $\log(1+x) \leq x$ to get:

$$\log \prod_{j=1}^n (1 + 2C_j) = \sum_{j=1}^n \log(1 + 2C_j) \leq \sum_{j=1}^n 2C_j,$$

hence,

$$\prod_{j=1}^{n}(1 + 2C_j) \le \exp\left(2\sum_{j=1}^{n}C_j\right).$$

To bound the sum $\sum_{j=1}^{\infty}C_j$, we split the numerator:

$$\sum_{j=1}^{\infty}C_j = \frac{1+\eta}{c}\sum_{j=1}^{\infty}\frac{\ln j}{j^{1+\eta}} + \frac{\ln(c)+7}{c}\sum_{j=1}^{\infty}\frac{1}{j^{1+\eta}}.$$

We use the known bounds:

$$\sum_{j=1}^{\infty}\frac{1}{j^{1+\eta}} \le 1 + \int_{1}^{\infty}\frac{1}{t^{1+\eta}}\mathrm{d}t = 1 + \frac{1}{\eta}, \quad \sum_{j=2}^{\infty}\frac{\ln j}{j^{1+\eta}} \le \int_{1}^{\infty}\frac{\ln t}{t^{1+\eta}}\mathrm{d}t = \frac{1}{\eta^2},$$

which gives:

$$\sum_{j=1}^{\infty}C_j \le \frac{1+\eta}{c\eta^2} + \frac{(\ln(c)+7)}{c}\left(1+\frac{1}{\eta}\right)$$

$$= \frac{1+\eta}{c\eta^2}\left(1 + \eta\cdot(\ln(c)+7)\right) =: S_{\eta,c}.$$

Then we have:

$$\prod_{j=1}^{n}(1+2C_j) \le \exp\left(2S_{\eta,c}\right), \quad \sum_{i=1}^{n}C_i\prod_{j=i+1}^{n}(1+2C_j) \le S_{\eta,c}\exp(2S_{\eta,c}).$$

Plugging these into the expression for $w_n$ yields the final bound:

$$w_n \le \exp(2S_{\eta,c})\left(w_0 + AS_{\eta,c}\right).$$

$\square$

# B  Controlling $\sigma \mapsto x_\sigma^\star$

The goal of this appendix is to show that the minimiser $x_\sigma^\star$ is Lipschitz-continuous with respect to $\sigma^2$. To establish this, we need to control how the objective function $F_\sigma$ evolves as $\sigma$ changes. A natural way to approach this is through a PDE perspective, since the smoothed density $p_\sigma$ satisfies the heat equation. This connection allows us to describe how $p_\sigma$, its logarithm, and its gradient (i.e., the score function) evolve with respect to $\sigma^2$.

Throughout this appendix, we use the following notation for differential operators acting on functions $f : \mathbb{R}^d \to \mathbb{R}$:

- $\nabla f$ denotes the gradient of $f$, a vector in $\mathbb{R}^d$,
- $\nabla^2 f$ denotes the Hessian of $f$, a $d \times d$ matrix of second-order partial derivatives,
- $\nabla^3 f$ denotes the third-order derivative tensor of $f$, a rank-3 tensor in $\mathbb{R}^{d \times d \times d}$,
- $\Delta f = \mathrm{tr}(\nabla^2 f)$ denotes the Laplacian of $f$.

The first lemma provides several PDEs satisfied by $p_\sigma$, $\ln p_\sigma$, and the score function $\nabla \ln p_\sigma$.

**Lemma 3.** *Let $p(x)$ be a probability density and denote by $p_\sigma(x)$ its convolution with an isotropic centered Gaussian of variance $\sigma^2$. For $\sigma > 0$, it holds that $p_\sigma(x) > 0$ for all $x \in \mathbb{R}^d$ and $p_\sigma$ follows the heat equation:*

$$\frac{\partial p_\sigma}{\partial \sigma^2} = \frac{1}{2} \Delta p_\sigma.$$

*Moreover, $-\ln p_\sigma$ follows the following partial differential equation:*

$$\frac{\partial \ln p_\sigma}{\partial \sigma^2} = \frac{1}{2}(\Delta \ln p_\sigma + \|\nabla \ln p_\sigma\|^2).$$

*Taking the gradient in the previous equation we get that the score functions follow:*

$$\frac{\partial \nabla \ln p_\sigma(x)}{\partial \sigma^2} = \frac{1}{2}\big[\nabla \Delta \ln p_\sigma(x) + 2[\nabla^2 \ln p_\sigma(x)]\nabla \ln p_\sigma(x)\big]$$

*Proof.* Standard results (see, e.g., [Evans, 2022, Chapter 2]) guarantee that $(\sigma, x) \mapsto p_\sigma(x)$ is $C^\infty$ on $\mathbb{R}_+^\star \times \mathbb{R}^d$ and satisfies the heat equation:

$$\frac{\partial p_\sigma}{\partial \sigma^2} = \frac{1}{2} \Delta p_\sigma.$$

By differentiating $\ln p_\sigma$ w.r.t. $\sigma^2$ and using the above, we directly have:

$$\frac{\partial \ln p_\sigma}{\partial \sigma^2} = \frac{1}{2} \frac{\Delta p_\sigma}{p_\sigma},$$

To get the PDE satisfied by $\ln p_\sigma$ notice that:

$$\Delta \ln p_\sigma = \frac{\Delta p_\sigma}{p_\sigma} - \|\nabla \ln p_\sigma\|^2,$$

Using both equation above directly yields:

$$\frac{\partial \ln p_\sigma}{\partial \sigma^2} = \frac{1}{2}(\Delta \ln p_\sigma + \|\nabla \ln p_\sigma\|^2).$$

Taking the gradient in the above identity leads to the last partial differential equation of the Lemma and concludes the proof. $\qquad\square$

This next lemma justifies the use of smoothed gradient descent by confirming that, as the smoothing parameter $\sigma \to 0$, the minimisers of the smoothed objectives $F_\sigma$ converge to the minimiser of the original (non-smoothed) objective $F$. In other words, the limit of the smoothed minimisers coincides with the proximal point we ultimately aim to recover.

**Lemma 4.** *Recall that we define*

$$F(x) := \frac{1}{2}\|y - x\|^2 - \tau \ln p(x) \quad \text{and} \quad F_\sigma(x) := \frac{1}{2}\|y - x\|^2 - \tau \ln p_\sigma(x).$$

*Recall that* $\mathrm{prox}_{-\tau \ln p}(y) := \underset{x \in \mathbb{R}^d}{\arg\min}\, F(x)$ *and that* $\mathrm{prox}_{-\tau \ln p_\sigma}(y) := \underset{x \in \mathbb{R}^d}{\arg\min}\, F_\sigma(x)$. *It holds that*

$$\mathrm{prox}_{-\tau \ln p_\sigma}(y) \underset{\sigma \to 0}{\to} \mathrm{prox}_{-\tau \ln p}(y).$$

*Proof.* Let $K$ be a compact set, since $p$ is continuous and $p(x) > 0$ on $K$ (Assumption 1), we have that there exists $a > 0$ such that $\inf_{x \in K} p(x) \geq a$. Now since $p$ is Lipschitz continuous on $K$, Theorem 2 in Nesterov and Spokoiny [2017] ensures that $\sup_{x \in K} |p_\sigma(x) - p(x)| \underset{\sigma \to 0}{\longrightarrow} 0$. Therefore for $\sigma$ small enough $\inf_{x \in K} p_\sigma(x) \geq a/2$ and from standard inequalities on the logarithm:

$$|\ln(p_\sigma(x)) - \ln(p(x))| \leq \frac{|p_\sigma(x) - p(x)|}{\min(p_\sigma(x), p(x))} \leq \frac{2}{a}|p_\sigma(x) - p(x)|.$$

Therefore $\sup_{x \in K} |\ln(p_\sigma(x)) - \ln(p(x))| \underset{\sigma \to 0}{\longrightarrow} 0$ on all compact sets $K$, and trivially:

$$\sup_{x \in K} |F_\sigma(x) - F(x)| \underset{\sigma \to 0}{\longrightarrow} 0.$$

To ease notations, let $x_\sigma^\star$ be the minimiser of $F_\sigma$ and $x^\star$ that of $F$. Note that such minimisers exist and are unique since $F_\sigma$ and $F$ are strongly convex by Proposition 4. Consider the values $F_\sigma(x^\star)$. By optimality of $x_\sigma^\star$ we know that $F_\sigma(x_\sigma^\star) \leq F_\sigma(x^\star)$. Moreover, since $F_\sigma \to F$ uniformly on compact sets, we have $F_\sigma(x^\star) \to F(x^\star)$, so in particular, the sequence $(F_\sigma(x_\sigma^\star))$ is uniformly bounded above:

$$F_\sigma(x_\sigma^\star) \leq F_\sigma(x^\star) \leq F(x^\star) + 1,$$

for $\sigma$ small enough. Now, assume that $\|x_\sigma^\star\| \to \infty$ along some sequence. Since the functions $F_\sigma$ are all 1-strongly convex, they can all be lower bounded by the same quadratic and we would have $F_\sigma(x_\sigma^\star) \to \infty$, contradicting the bound above. Therefore, the sequence $(x_\sigma^\star)_{\sigma^2 \in (0,\tau]}$ is bounded, and thus contained in a fixed compact set $K \subset \mathbb{R}^d$.

Since $F_\sigma \to F$ uniformly on $K$, any cluster point $x_\infty$ of $(x_\sigma^\star)$ satisfies

$$F(x_\infty) = \lim_{\sigma \to 0} F_\sigma(x_\sigma^\star) \leq \lim_{\sigma \to 0} F_\sigma(x^\star) = F(x^\star).$$

Therefore, by uniqueness of the minimiser of $F$, it must be that $x_\infty = x^\star$ so that $x_\sigma^\star \underset{\sigma \to 0}{\longrightarrow} x^\star$. $\quad\square$

The next proposition establishes the existence and smoothness of the solution path $x_\sigma^\star$ as a function of $\sigma$.

**Proposition 7** (Existence of the smooth solution path). *Recall that*

$$F_\sigma(x) := \frac{1}{2}\|y - x\|^2 - \tau \ln p_\sigma(x).$$

*Denote by $x_\sigma^\star$ the minimiser of $F_\sigma$ for any $\sigma > 0$. Then $\sigma^2 \mapsto x_\sigma$ is continuously differentiable on $(0, \tau]$ and satisfies the following ordinary differential equation:*

$$\frac{\mathrm{d}x_\sigma^\star}{\mathrm{d}\sigma^2} =: \dot{x}_\sigma^\star = -\nabla^2 F_\sigma(x_\sigma^\star)^{-1} \partial_{\sigma^2} \nabla F_\sigma(x_\sigma^\star).$$

*Proof.* By smoothness of the solution of the heat equation (see, e.g., [Evans, 2022, Chapter 2]), we have that $x \mapsto F_\sigma(x)$ is differentiable for any $\sigma > 0$ and $(\sigma^2, x) \mapsto \nabla_x F_\sigma(x)$ is jointly differentiable on $\mathbb{R}_+^\star \times \mathbb{R}^d$. Then, by Proposition 4, we have that the Hessian $\nabla^2 F_\sigma(x)$ is invertible and satisfies: $\nabla^2 F_\sigma(x) \succeq I_d$. We can then apply the implicit function theorem, which guarantees the existence of a unique solution path $\sigma^2 \mapsto x_\sigma^\star$ to the implicit equation: $\nabla F_\sigma(x_\sigma^\star) = 0$ that is differentiable on $(0, \tau]$. By strong convexity of $F_\sigma$, this solution path coincides with the minimisers of $F_\sigma$ for all $\sigma > 0$. The ODE followed by $\sigma^2 \to x_\sigma^\star$ is obtained by taking the derivative with respect to $\sigma^2$ of the identity $\nabla F_\sigma(x_\sigma^\star) = 0$. $\quad\square$

**Proposition 8** (Bound on the solutions). *Let $x_\sigma^\star := \arg\min_{x \in \mathbb{R}^d} F_\sigma(x)$, then for $\sigma^2 \leq \tau$ it holds that*

$$\|x_\sigma^\star - y\| \leq \|y - \text{prox}_{-\tau \ln p}(y)\| + \frac{1}{2}\tau^2 M\sqrt{d}$$

*Proof.* Let use write $\dot{x}_\sigma^\star = \frac{\mathrm{d}x_\sigma^\star}{\mathrm{d}\sigma^2}$ (note that the derivative is with respect to $\sigma^2$ and not $\sigma$). From Proposition 7, we have that $x_\sigma^\star$ follows the differential equation:

$$\begin{aligned}
\dot{x}_\sigma^\star &= -\nabla^2 F_\sigma(x_\sigma^\star)^{-1}\partial_{\sigma^2}\nabla F_\sigma(x_\sigma^\star) \\
&= \tau \nabla^2 F_\sigma(x_\sigma^\star)^{-1}\partial_{\sigma^2}\nabla \ln p_\sigma(x_\sigma^\star) \\
&= \frac{1}{2}[-\nabla^2 \ln p_\sigma(x_\sigma^\star) + \frac{1}{\tau}I_d]^{-1}[\nabla\Delta \ln p_\sigma(x_\sigma^\star) + 2[\nabla^2 \ln p_\sigma(x_\sigma^\star)]\nabla \ln p_\sigma(x_\sigma^\star)] \quad (9)
\end{aligned}$$

where the last equality follows from Lemma 3. Furthermore, recalling the optimality condition satisfied by $x_\sigma^\star$, i.e.: $\nabla \ln p_\sigma(x_\sigma^\star) = \frac{1}{\tau}(x_\sigma^\star - y)$, if follows that:

$$\dot{x}_\sigma^\star = -\frac{1}{2\tau}Q_\sigma(x_\sigma^\star - y) + B_\sigma, \quad (10)$$

where the matrix $Q_\sigma$ and vector $B_\sigma$ are given by:

$$Q_\sigma := -[-\nabla^2 \ln p_\sigma(x_\sigma^\star) + \frac{1}{\tau}I_d]^{-1}\nabla^2 \ln p_\sigma(x_\sigma^\star) \succeq 0 \quad (11)$$

$$B_\sigma := \frac{1}{2}[-\nabla^2 \ln p_\sigma(x_\sigma^\star) + \frac{1}{\tau}I_d]^{-1}\Delta\nabla \ln p_\sigma(x_\sigma^\star). \quad (12)$$

Here, the matrix $Q_\sigma$ is positive semi-definite since $-\nabla^2 \ln p_\sigma(x)$ is positive by Proposition 4. Now from eq. (10), we get:

$$\begin{aligned}
\frac{1}{2}\frac{\mathrm{d}\|x_\sigma^\star - y\|^2}{\mathrm{d}\sigma^2} &= \langle \dot{x}_\sigma^\star, x_\sigma^\star - y\rangle \\
&= -\frac{1}{2\tau}\|x_\sigma^\star - y\|_{Q_\sigma}^2 + \langle B_\sigma, x_\sigma^\star - y\rangle \\
&\leq \langle B_\sigma, x_\sigma^\star - y\rangle \\
&\leq \|B_\sigma\|\|x_\sigma^\star - y\|.
\end{aligned}$$

From the upperbound $\|\nabla\Delta \log p_\sigma(x_\sigma^\star)\| \leq M\sqrt{d}$ which follows from Lemma 5, we directly have that $\|B_\sigma\| \leq \frac{\tau}{2}M\sqrt{d}$. Injecting this bound in the above inequality and dividing both sides by $\|x_\sigma^\star - y\|$ yields:

$$\frac{\mathrm{d}\|x_\sigma^\star - y\|}{\mathrm{d}\sigma^2} \leq \frac{\tau}{2}M\sqrt{d}.$$

Integrating of the above inequality from 0 to $\sigma^2$, using that $\lim_{\sigma \to 0} x_\sigma^\star = \text{prox}_{-\tau \ln p}(y)$ from Lemma 4, we get:

$$\begin{aligned}
\|x_\sigma^\star - y\| &\leq \|y - \text{prox}_{-\tau \ln p}(y)\| + \frac{1}{2}\sigma^2\tau M\sqrt{d} \\
&\leq \|y - \text{prox}_{-\tau \ln p}(y)\| + \frac{1}{2}\tau^2 M\sqrt{d},
\end{aligned}$$

where the last inequality is since we consider $\sigma^2 \leq \tau$. $\qquad \square$

**Proposition 9** (Lipschitz continuity of $\sigma^2 \mapsto x_\sigma^\star$). *Let $x_\sigma^\star := \arg\min_{x \in \mathbb{R}^d} F_\sigma(x)$, then for $\sigma_2^2 \leq \sigma_1^2 \leq \tau$, it holds that:*

$$\|x_{\sigma_1}^\star - x_{\sigma_2}^\star\| \leq (\sigma_1^2 - \sigma_2^2)\left[\frac{1}{\tau}\|y - \text{prox}_{-\tau \ln p}(y)\| + \tau M\sqrt{d}\right],$$

*And taking $\sigma_2 \to 0$ in the above inequality:*

$$\|x_\sigma^\star - \text{prox}_{-\tau \ln p}(y)\| \leq \sigma^2\left[\frac{1}{\tau}\|y - \text{prox}_{-\tau \ln p}(y)\| + \tau M\sqrt{d}\right],$$

*Proof.* Recall from Equation (9):

$$\dot{x}_\sigma^\star = \frac{1}{2}[-\nabla^2 \ln p_\sigma(x_\sigma^\star) + \frac{1}{\tau}I_d]^{-1}[\nabla\Delta \ln p_\sigma(x_\sigma^\star) + 2[\nabla^2 \ln p_\sigma(x_\sigma^\star)]\nabla \ln p_\sigma(x_\sigma^\star)]$$

Now, by Proposition 4, we have that $-\nabla^2 \ln p_\sigma(x) \succeq 0$, and a spectral norm bound on the inverse yields:

$$\|[-\nabla^2 \ln p_\sigma(x_\sigma^\star) + \frac{1}{\tau}I_d]^{-1}\nabla\Delta \ln p_\sigma(x_\sigma^\star)\| \leq \tau\|\nabla\Delta \ln p_\sigma(x_\sigma^\star)\|$$

and:

$$\|[-\nabla^2 \ln p_\sigma(x_\sigma^\star) + \frac{1}{\tau}I_d]^{-1}[\nabla^2 \ln p_\sigma(x_\sigma^\star)]\nabla \ln p_\sigma(x_\sigma^\star)\| \leq \|\nabla \ln p_\sigma(x_\sigma^\star)\|.$$

Putting things together we obtain that:

$$\|\dot{x}_\sigma^\star\| \leq \|\nabla \ln p_\sigma(x_\sigma^\star)\| + \frac{\tau}{2}\|\nabla\Delta \ln p_\sigma(x_\sigma^\star)\| \tag{13}$$

$$\leq \|\nabla \ln p_\sigma(x_\sigma^\star)\| + \frac{\tau}{2}M\sqrt{d}, \tag{14}$$

where the second inequality is due to Lemma 5. Now recall that the optimality condition which define $x_\sigma^\star$ is $\nabla \ln p_\sigma(x_\sigma^\star) = \frac{1}{\tau}(x_\sigma^\star - y)$. Plugging this equality in the upperbound we get that:

$$\|\dot{x}_\sigma^\star\| \leq \frac{1}{\tau}\|y - x_\sigma^\star\| + \frac{\tau}{2}M\sqrt{d}$$

$$\leq \frac{1}{\tau}\|y - \operatorname{prox}_{-\tau \ln p}(y)\| + \tau M\sqrt{d},$$

where the last inequality is due to Proposition 8.

From here it suffices to notice that, for $\sigma_1 \geq \sigma_2 > 0$:

$$\|x_{\sigma_1}^\star - x_{\sigma_2}^\star\| = \left\|\int_{\sigma_1^2}^{\sigma_2^2} \dot{x}_\sigma^\star d\sigma^2\right\|$$

$$\leq \int_{\sigma_1^2}^{\sigma_2^2} \|\dot{x}_\sigma^\star\|d\sigma^2$$

$$\leq (\sigma_1^2 - \sigma_2^2)\left[\frac{1}{\tau}\|y - \operatorname{prox}_{-\tau \ln p}(y)\| + \tau M\sqrt{d}\right],$$

which proves the first statement. The second follows from the fact that $x_{\sigma_2}^\star \xrightarrow[\sigma_2 \to 0]{} \operatorname{prox}_{-\tau \ln p}(y)$ by Lemma 4. □

This last result is the most technical lemma in this work. It establishes that the third derivative of the smoothed log-density $\ln p_\sigma$ can be uniformly controlled—independently of $\sigma$. This regularity bound is essential for tracking how the minimisers $x_\sigma^\star$ evolve as $\sigma$ varies.

**Lemma 5.** *For all $\sigma \geq 0$, it holds that $\sup_{x \in \mathbb{R}^d} \|\nabla\Delta \ln p_\sigma(x)\| \leq \sqrt{d}M$.*

**We would like to emphasise again that the following proof is entirely based on the computations and insights that Filippo Santambrogio generously shared with us in response to an email we sent asking for ideas on how to approach this result. The proof is technical and relies on several surprising simplifications that Filippo identified.**

*Proof.* To simplify notations, throughout the proof we let $t := \sigma^2$ and let $V(t, x) := -\ln p_{\sqrt{t}}(x) = \ln p_\sigma(x)$ correspond to the convex potential associated to $p_\sigma$. The proof first relies on showing that $\|\nabla^3 V(t, x)\|$ must be maximal for $t = 0$.

**Establishing a parabolic inequality for $\|\nabla^3 V(t,x)\|$.** From Lemma 3, we have that the potential $V$ follows the following PDE:

$$\partial_t V = \frac{1}{2}(\Delta V - \|\nabla V\|^2).$$

For $i, j, k \in [d]$, we let $w_{ijk} := \partial_{ijk} V$, which therefore follows:

$$\partial_t w_{ijk} = \frac{1}{2}(\Delta w_{ijk} - \partial_{ijk}\|\nabla V\|^2).$$

Now let $u_{ijk} = w_{ijk}^2$, multiplying the previous equation by $w_{ijk}$ we get:

$$\begin{aligned}
\partial_t u_{ijk} &= w_{ijk}\Delta w_{ijk} - w_{ijk}\partial_{ijk}\|\nabla V\|^2 \\
&= \frac{1}{2}\left(\Delta u_{ijk} - (\Delta w_{ijk})^2\right) - w_{ijk}\partial_{ijk}\|\nabla V\|^2 \\
&\leq \frac{1}{2}\Delta u_{ijk} - w_{ijk}\partial_{ijk}\|\nabla V\|^2
\end{aligned}$$

Summing over $i, j, k$ and letting $S(t,x) := \|\nabla^3 V(t,x)\|^2 = \sum_{ijk} u_{ijk}$, we have that:

$$\partial_t S \leq \frac{1}{2}\Delta S - \sum_{ijk} w_{ijk}\partial_{ijk}\|\nabla V\|^2$$

It remains to control the last term in the inequality. Since $\|\nabla V\|^2 = \sum_\ell (\partial_l V)^2$, taking the third derivative with respect to $i, j, k$ we get that:

$$\begin{aligned}
\partial_{ijk}\|\nabla V\|^2 &= 2\sum_\ell \partial_l V \cdot \partial_{ijkl}V + \partial_{jkl}V \cdot \partial_{il}V + \partial_{ikl}V \cdot \partial_{jl}V + \partial_{ijl}V \cdot \partial_{kl}V \\
&= 2\langle \nabla V, \nabla w_{ijk}\rangle + 2\sum_\ell w_{jkl} \cdot \partial_{il}V + w_{ikl} \cdot \partial_{jl}V + w_{ijl} \cdot \partial_{kl}V.
\end{aligned}$$

Multiplying the equality by $w_{ijk}$ and summing over $i, j, k$ we get:

$$\sum_{ijk} w_{ijk}\partial_{ijk}\|\nabla V\|^2 = \langle \nabla V, \nabla S\rangle + 2\sum_{ijk\ell} w_{ijk}w_{jkl} \cdot \partial_{il}V + w_{ijk}w_{ikl} \cdot \partial_{jl}V + w_{ijk}w_{ijl} \cdot \partial_{kl}V.$$

However notice that from the convexity of $V(\sigma, \cdot)$ for all $\sigma \geq 0$, we get that:

$$\sum_{jk} \Big( \underbrace{\sum_{i\ell} w_{ijk}w_{jkl} \cdot \partial_{il}V}_{\geq 0} \Big) \geq 0,$$

which implies that the function $S(t,x) := \|\nabla^3 V(t,x)\|^2$ satisfies the following parabolic inequality

$$\partial_t S \leq \frac{1}{2}\Delta S - \langle \nabla V, \nabla S\rangle. \tag{15}$$

**Proving that $S$ is maximal for $t = 0$.** To prove that $S$ must attain its maximum for $t = 0$, let us fix $t_1 > 0$ and for $t \in [0, t_1]$, we let $\tilde{S}(t,x) = S(t_1 - t, x)$ and $\tilde{V}(t,x) = V(t_1 - t, x)$ correspond to the "reversed time" counterparts of $S$ and $V$. Adapting Equation (15), the parabolic inequality satisfied by $\tilde{S}$ is:

$$\partial_t \tilde{S} \geq -\frac{1}{2}\Delta \tilde{S} + \langle \nabla \tilde{V}, \nabla \tilde{S}\rangle. \tag{16}$$

For $t \in [0, t_1]$, we now consider the following stochastic differential equation:

$$\mathrm{d}X_t = -\nabla \tilde{V}(t, X_t)\mathrm{d}t + \mathrm{d}B_t, \tag{17}$$

initialised at $X_{t=0} = x_0$ for some $x_0 \in \mathbb{R}^d$. From Lemma 6, we are guaranteed the existence and uniqueness of a strong solution to this stochastic differential equation over $[0, t_1]$. We can then apply the Itô formula to $\tilde{S}(t, X_t)$:

$$\begin{aligned}
\mathrm{d}\tilde{S}(t, X_t) &= \partial_t \tilde{S}(t, X_t)\,\mathrm{d}t + \langle \nabla \tilde{S}(t, X_t), \mathrm{d}X_t\rangle + \frac{1}{2}\Delta \tilde{S}(t, X_t)\mathrm{d}t \\
&= \partial_t \tilde{S}(t, X_t)\,\mathrm{d}t - \langle \nabla \tilde{S}(t, X_t), \nabla \tilde{V}(t, X_t)\rangle\mathrm{d}t + \frac{1}{2}\Delta \tilde{S}(t, X_t)\mathrm{d}t + \langle \nabla \tilde{S}(t, X_t), \mathrm{d}B_t\rangle \\
&\geq \langle \nabla \tilde{S}(t, X_t), \mathrm{d}B_t\rangle,
\end{aligned}$$

where the last inequality is due to the parabolic inequality on $\tilde{S}$ from eq. (16). Now integrating from $t = 0$ to $t = t_1$ we obtain:

$$\tilde{S}(t_1, X_{t_1}) \geq \tilde{S}(0, X_{t=0}) + \int_0^{t_1} \langle \nabla \tilde{S}(t, X_t), \mathrm{d}B_t \rangle$$

$$= \tilde{S}(0, x_0) + \int_0^{t_1} \langle \nabla \tilde{S}(t, X_t), \mathrm{d}B_t \rangle.$$

Since the expectation of the stochastic integral is 0, and recalling that $\tilde{S}(t, x) = S(t_1 - t, x)$, we obtain:

$$\mathbb{E}[S(0, X_{t_1})] = \mathbb{E}[\tilde{S}(t_1, X_{t_1})] \geq \tilde{S}(0, x_0) = S(t_1, x_0).$$

It remains to use that $\sup_x S(0, x) < \infty$ from Assumption 2 to obtain that:

$$\sup_{x \in \mathbb{R}^d} S(0, x) \geq \mathbb{E}[S(0, X_{t_1})] \geq S(t_1, x_0).$$

Since this inequality holds for all $x_0 \in \mathbb{R}^d$ and $t_1 > 0$, we get that:

$$\sup_{x \in \mathbb{R}^d} S(t, x) \leq \sup_{x \in \mathbb{R}^d} S(0, x), \quad \forall t \geq 0.$$

Therefore, recalling that $S(t, x) := \|\nabla^3 V(t, x)\|^2 = \|\nabla^3 \ln p_{\sqrt{t}}(x)\|^2$, we finally have that for all $\sigma \geq 0$:

$$\sup_{x \in \mathbb{R}^d} \|\nabla^3 \ln p_\sigma(x)\| \leq \sup_{x \in \mathbb{R}^d} \|\nabla^3 \ln p(x)\|.$$

**From $\|\nabla^3\|$ to $\|\nabla\Delta\|$.** From the Cauchy-Schwartz inequality, one gets:

$$\|\nabla \Delta f\|^2 = \sum_{i=1}^d \left( \sum_{j=1}^d \partial_{ijj} f \right)^2 \leq d \sum_{i,j=1}^d (\partial_{ijj} f)^2 \leq d \sum_{i,j,k=1}^d (\partial_{ijk} f)^2 = d\|\nabla^3 f\|^2,$$

which concludes the proof.

$\square$

**Lemma 6.** *For a horizon time $t_1 > 0$, let $\tilde{V}(t, x) = -\ln p_{\sqrt{t_1 - t}}(x)$ denote the backward-time log-density defined over $[0, t_1] \times \mathbb{R}^d$. Then for all initialisation $X_{t=0} = x_0 \in \mathbb{R}^d$, the stochastic differential equation defined in Equation (17) which we recall here:*

$$\mathrm{d}X_t = -\nabla \tilde{V}(t, X_t)\mathrm{d}t + \mathrm{d}B_t,$$

*has a unique strong solution over $[0, t_1]$.*

*Proof.* From Proposition 4 we have for all $x \in \mathbb{R}^d$:

$$0 \preceq \nabla^2 V(t, x) = -\nabla^2 \ln p_{\sqrt{t}}(x) \preceq \frac{1}{t} I_d.$$

Therefore $\tilde{V}(t, x) := V(t_1 - t, x)$ satisfies:

$$0 \preceq \nabla^2 \tilde{V}(t, x) \preceq \frac{1}{t_1 - t} \cdot I_d.$$

This entails that for all $\varepsilon > 0$, $\nabla \tilde{V}$ is globally Lipschitz for $t \in [0, t_1 - \varepsilon]$:

$$\|\nabla \tilde{V}(t, x) - \nabla \tilde{V}(t, x')\| \leq \frac{1}{\varepsilon} \|x - x'\|,$$

which ensures the existence of a unique strong solution over $[0, t_1 - \varepsilon]$ (see e.g. Theorem 5.2.1 in Oksendal [2013]) and hence over $[0, t_1)$. It remains to show that $X_t$ does not blow up as $t \to t_1^-$.

**Proving that $X_t$ is bounded over $[0, t_1)$.** To do so, we consider the Lyapunov $\frac{1}{2}\|X_t - x_0\|^2$, for which the Itô formula provides that:

$$\frac{1}{2}\mathrm{d}\|X_t - x_0\|^2 = \langle \mathrm{d}X_t, X_t - x_0 \rangle + \frac{d}{2}\mathrm{d}t$$

$$= \langle \nabla \tilde{V}(t, X_t), x_0 - X_t \rangle \mathrm{d}t + \frac{d}{2}\mathrm{d}t + \langle \mathrm{d}B_t, X_t - x_0 \rangle.$$

Now recall that for all $t$, the function $x \mapsto V(t, x)$ is convex (Proposition 4) and hence we have the inequality $\langle \nabla V(t, x'), x - x' \rangle \le V(t, x) - V(t, x')$, which leads to:

$$\frac{1}{2}\mathrm{d}\|X_t - x_0\|^2 \le (\tilde{V}(t, x_0) - \tilde{V}(t, X_t))\mathrm{d}t + \frac{d}{2}\mathrm{d}t + \langle \mathrm{d}B_t, X_t - x_0 \rangle.$$

Recalling the integral definition of $p_\sigma$ as $p_\sigma(x) = \int_{\mathbb{R}^d} p(z)\phi_\sigma(x - z)\mathrm{d}z$, where $\phi_\sigma$ denotes gaussian density function of variance $\sigma^2 = t$, we have that $\sup_x p_\sigma(x) \le p_{\max} := \sup_x p(x)$ as well as $\inf_{\sigma \in [0, t_1]} p_\sigma(x_0) =: p_{\min}(x_0) > 0$ (since $p$ is assumed strictly positive over $\mathbb{R}^d$ from Assumption 1). Therefore

$$\mathrm{d}\|X_t - x_0\|^2 \le C\mathrm{d}t + 2\langle X_t - x_0, \mathrm{d}B_t \rangle,$$

with $C = 2\ln(p_{\max}/p_{\min}(x_0)) + d$. Now integrating from 0 to $t < t_1$ we obtain:

$$\|X_t - x_0\|^2 \le Ct + 2\int_0^{t'} \langle X_{t'} - x_0, \mathrm{d}B_{t'} \rangle$$

$$\le Ct + M_t, \tag{18}$$

where $M_t := 2\int_0^t \langle X_{t'} - x_0, \mathrm{d}B_{t'} \rangle$ is a continuous-time martingale.

**Bounding $M_t$ over $[0, t_1)$** Taking the expectation in the last inequality we get:

$$\mathbb{E}[\|X_t - x_0\|^2] \le Ct \le Ct_1.$$

Now notice that due to the Itô isometry, we have that:

$$\mathbb{E}[M_t^2] = 4\mathbb{E}\left[\int_0^t \|X_{t'} - x_0\|^2 \mathrm{d}t'\right] = 4\int_0^t \mathbb{E}[\|X_{t'} - x_0\|^2]\mathrm{d}t' \le 4Ct_1^2.$$

We now apply Doob's martingale inequality to the process $M_t^2$:

$$\mathbb{P}\left(\sup_{t' \le t} M_{t'}^2 \ge A^2\right) \le \frac{\mathbb{E}[M_t^2]}{A^2} \le \frac{4Ct_1^2}{A^2}.$$

And since

$$\left\{\sup_{t' < t_1} M_{t'}^2 \ge A^2\right\} = \bigcup_{n \ge 1}\left\{\sup_{t' < t_1 - \frac{1}{n}} M_{t'}^2 \ge A^2\right\},$$

where the sequence of events are monotonically increasing, we obtain that:

$$\mathbb{P}\left(\sup_{t' < t_1} M_{t'}^2 \ge A^2\right) = \lim_{n \to \infty} \mathbb{P}\left(\sup_{t' < t_1 - \frac{1}{n}} M_{t'}^2 \ge A^2\right) \le \frac{4Ct_1^2}{A^2}.$$

Therefore $\lim_{A \to \infty} \mathbb{P}\left(\sup_{t < t_1} M_t^2 \ge A^2\right) = 0$ which translates into:

$$\mathbb{P}\left(\sup_{t < t_1} M_t < \infty\right) = 1.$$

Due to inequality 18, this means that the trajectories $(X_t(\omega))_{t \in [0, t_1)}$ are bounded for almost all $\omega$. Therefore, due to the continuity of $\nabla \tilde{V}(t, x)$ over $\mathbb{R} \times \mathbb{R}^d$, the path $t \mapsto \tilde{V}(t, X_t(\omega))$ is bounded on $[0, t_1)$. Hence, for almost all $\omega$,

$$X_t(\omega) = x_0 - \int_0^t \nabla \tilde{V}(t', X_{t'}(\omega))\mathrm{d}t' + B_t(\omega)$$

must admit a limit when $t \to t_1^-$. Hence $X_t$ extends continuously to $t = t_1$ and $X_{t_1}(\omega)$ still satisfies the integral form of the SDE. Hence a strong solution exists on the whole interval $[0, t_1]$. Unicity over $[0, t_1]$ follows from unicity over $[0, t_1)$ and taking the limit in $t_1^-$.

$\square$

