# OpenReview forum: "MAP Estimation with Denoisers: Convergence Rates and Guarantees"
_NeurIPS.cc/2025/Conference — NeurIPS 2025 poster_

### Official Review · Reviewer_3djj · 2025-06-28

**Clarity:** 3
**Significance:** 2
**Originality:** 3
**Rating:** 4
**Confidence:** 2

**Summary:**

Denoisers are often used as priors to solve inverse problems, typically by formulating the problem as a maximum a posteriori (MAP) estimation with a negative log-prior. While pre-trained denoisers commonly approximate this prior, there's generally no theoretical basis for this practice. This paper addresses that gap, demonstrating that a simple, practical algorithm (similar to the commonly used) provably converges to the proximal operator under a log-concavity assumption of the prior. The work also offers novel interpretations of the optimization problem as a gradient descent on a smoothed proximal objective.

**Questions:**

I have one questions that could help me clarify my understanding of the work.

My main question is regarding Algorithm 1. The inner-loop that does the MMSE averaging is with fixed weights and noise levels. This means this loop is solving a fixed point. I don't see the connection of this fixed point with the MMSE averaging equation where $\alpha_k$ evolves at each iteration. Same with the connection with Cold Diffusion (where the weights evolve at every step).

I would also like the authors to comment on the significance. This is theoretic paper without any real numerical experiment. The theoretical result is interesting but still makes some assumptions that are not generally hold. This limits the impact.

Besides this question, I was wondering if there's a way to push the discretization and the numerical Algorithm 1, into an ODE formulation  (e.g, $\tau \to 0$). This could help with the understanding/intuition and also to develop connections to other works (diffusion, PF-ODE). See for example Appendix B in Delbracio and Milanfar 2023 where they analyze the case of a Gaussian prior.

Notation:
 In Algorithm 1. $k$ represents the iteration on the outer loop, but within the text $k$ represents a different thing (see e.g., Theorem 2). This complicates the understanding.

**Ethical Concerns:**

["NO or VERY MINOR ethics concerns only"]

**Final Justification:**

This is an interesting theory paper, well written, and which drives connections to relevant work. The main limitations is that it doesn't show any real application of the theory, but the theoretical result is still interesting (though based on some strong assumptions). The paper had minor typos that are going to be corrected in the revised version. My recommendation thus is borderline accept.

**Limitations:**

Limitations are not explicitly stated but the paper discusses future work about extending the analysis to more realistic cases ( L343-L346).

**Paper Formatting Concerns:**

No paper formatting concerns.

**Quality:**

3

**Strengths And Weaknesses:**

Strengths:
- The paper is generally well written, the connections with relevant work as correctly presented in a nice manner.
- The paper introduces connections to other works (e.g., Cold Diffusion) which is something quite interesting.

Weaknesses:
 - The main weaknesses is that the paper doesn't show any real application of the theory. The theoretical result is interesting though, so it might be that there's no need for showing applications, but having these will make the paper much stronger.
- There are some minor presentation issues (repeated notation, for example "k" seems to mean something different in the main body of text than on the Algorithm 1.

---

> ### Author Rebuttal · Authors · 2025-07-27
>
> We thank the reviewer for their comments and thoughtful engagement with our work.
> The concerns expressed in the review are addressed below.
>
>  - **Algorithm 1.** Thanks for catching this, there were indeed some typos in Algorithm 1 (note that the analysis was done on the correct algorithm). The corrected algorithm uses a schedule for noise levels $\sigma_k$ and weights $\alpha_k$. We will also fix the overloading of variable $k$ for the iteration index in the main text which is confusing: we will use $i$ for the PGD algorithm loop and $k$ for the MMSE Averaging loop (it is fixed in the algorithm below). The correct algorithm looks like this:
>
>     $\\hat{x}^{(0)} \\gets y$
>
>     for $i = 0, 1, \\dots$ do
>
>     $\\quad \\hat{x}\_0^{(i+1)} = \\hat{x}^{(i)} - \\tau \\lambda \\nabla f(\\hat{x}^{(i)})$
>
>     $\\quad \\text{for}~k = 0, 1, \\dots, n_{i+1} - 1$ do
>
>     $\\qquad\\sigma\_k \\gets \\sqrt{\\frac{\\tau}{k + 1}}$
>
>     $\\qquad\\alpha_k \\gets \\frac{k+1}{k+2}$
>
>     $\\qquad\\hat{x}\_{k+1}^{(i+1)} \\gets \\alpha_k \\mathrm{MMSE}\_{\\sigma\_k}(\\hat{x}\_k^{(i+1)} ) + (1 - \\alpha\_k) \\hat{x}\_{0}^{(i+1)}$
>
>     $\\quad\\hat{x}^{(i+1)} \\gets \\hat{x}^{(i+1)}\_{n\_{i+1}}$
>
>  - **Continuous-time version.** It is indeed possible to derive a continuous-time analogue of our gradient descent recursion on the smoothed proximal objectives. Specifically, one can write
>     \\[ \\dot{x}_t = - \\gamma\_t \nabla F\_{\\sigma\_t}(x\_t) = - \gamma\_t (x\_t - y) + \\gamma\_t \\tau \\nabla \\ln p\_{\\sigma\_t}(x\_t) \\]
>
>     where $\\sigma_t$ and $\\gamma_t$ are time-dependent parameters.
>     Now assuming that $\\gamma_t = (1 + \\tau / \\sigma_t^2)^{-1}$, this can be equivalently rewritten as
>     \\[\\dot{x}\_t = - x\_t + \\gamma\_t y + (1 - \\gamma\_t) \\mathrm{MMSE}\_{\\sigma\_t}(x\_t) \\]
>     which would correspond to the continuous-time equivalent of our MMSE averaging sequence.
>     Assuming that $p(x)$ is a Gaussian (hence the MMSE has a closed form), and that $\sigma_t$ decreases as $\tau / t$, we can solve the ODE in closed form and show that it converges to the proximal solution.
>     However, while the ODEs have a similar shape to the one derived in Appendix B of *Delbracio & Milanfar (2023)*, the two algorithms have different objectives and are not entirely comparable: the proximal solution for us, samples from $p(x)$ for InDI. Of course the type of convergence proved in InDi is weaker than what we prove in that it has a) restrictive assumptions requiring Gaussian $p(x)$ and b) is based on a continuous limit which can't be achieved in practice, while ours is based on a discrete recursion.
>
>  - **Significance.** We agree that the assumptions (log-concavity, differentiability) do limit direct applicability. However, as with many theoretical papers, we believe this work serves as a first step toward understanding the behaviour of practical denoising-based schemes from a variational optimisation perspective, especially for a problem where no convergence rates have been proven so far. Besides, even under strong assumptions, establishing convergence to the *true* proximal operator and MAP estimator is highly non-trivial. We hope this theoretical foundation can guide future extensions to more general priors.
>
> - **Lack of experiments.** Our primary goal in this paper is to provide theoretical foundations for a class of heuristics already used in practice—such as in *Cold Diffusion*—which have demonstrated strong empirical performance. Nevertheless, we agree that a thorough empirical study is very valuable, and we believe it would hold as a separate work by itself.

---

> > ### Comment · Reviewer_3djj · 2025-08-04
> >
> > I appreciate the authors effort in answering the raised points. I believe this is an interesting paper, and the limitations are already discussed there. I'm happy to keep my (borderline) acceptance score.

---

### Official Review · Reviewer_vJDD · 2025-06-29

**Clarity:** 3
**Significance:** 1
**Originality:** 2
**Rating:** 4
**Confidence:** 4

**Summary:**

The submitted paper follows a plug-and-play (PnP) strategy [VBW 2013] to solve the maximum a posteriori (MAP) problem (MAP). In the strategy, the proximal gradient of the negative-log prior is replaced with the recursion (MMSE Averaging). The main results are Theorems 1 and 2. Theorem 1 claims that the MMSE averaging converges to the minimizer of the proximal objective (Proximal Objective) at the rate $O(1/k)$, with $k$ denoting the number of iterations in the MMSE averaging. Theorem 1 requires two technical assumptions: In Assumption 1, the prior density is assumed log-concave and strictly positive on $\mathbb{R}^{d}$, i.e. its support is equal to $\mathbb{R}^{d}$. Assumption 2 postulates the existence and boundedness of the third derivates of the log prior. Theorem 2 claims that approximate proximal gradient descent (PGD) obtained by plugging in the MMSE averaging converges to the true MAP solution at the rate $O(1/k)$ in a special sense presented in Theorem 2. A numerical visualization of the MMSE averaging is also presented with a two-dimensional Gaussian toy model.

**Questions:**

--Line 172:
$\ln p$ should be $-\ln p$ since $\ln p$ is not convex but concave.

--Proposition 1:
Is the weight not $\alpha_k = 1/(k+2)$ but $1 - \alpha_k = 1/(k+2)$? Please check it. If it is true, this typo appears in Theorem 1 and Algorithm 1 while it does not provide any changes in the theorems.

--Proposition 3 and Theorem 2:
Add the phrase “under Assumption 1” to Proposition 3. Theorem 2 is also in the same situation. Theorem 2 requires Assumptions 1 and 2 since Theorem 1 is used in its proof.

**Ethical Concerns:**

["NO or VERY MINOR ethics concerns only"]

**Final Justification:**

Authors' rebuttal has resolved my concern about the significance of the assumptions. Thus, I increase my score.

**Limitations:**

yes

**Paper Formatting Concerns:**

No problem

**Quality:**

3

**Strengths And Weaknesses:**

Originality:
The MMSE averaging itself is not novel. For instance, a similar algorithm can be found in [Feng etl al. 2024, Algorithm 1]. As pointed out in the submitted paper, the MMSE averaging is inspired by a diffusion model like [Bansal et al. 2023]. Algorithm 1 in the submitted paper is also in the same situation: It is PnP itself. Thus, the novel results of the submitted paper are Theorems 1 and 2.

[Feng etl al. 2024] T. Li, H. Feng, L. Wang, L. Zhu, Z. Xiong and H. Huang, "Stimulating Diffusion Model for Image Denoising via Adaptive Embedding and Ensembling," in IEEE Transactions on Pattern Analysis and Machine Intelligence, vol. 46, no. 12, pp. 8240-8257, Dec. 2024.

Significance:
The convergence results themselves in Theorems 1 and 2 are good. However, they require strong assumptions in Assumptions 1 and 2, such as the log-concavity, three times differentiability, and its boundedness for the prior density. For instance, the $\ell_1$ regularization (or the Laplace prior) is not differentiable at the origin. In my opinion, these strong assumptions exclude almost all priors requiring PnP. ReLU cannot be used in the denoising network. Thus, the practical significance of Theorems 1 and 2 is questionable.

Theorem 2 is weak from a different point of view. It only claims the average $k^{-1}\sum_{i=1}^{k}J(x_i)$ converges to the minimum of the MAP problem at the rate $O(1/k)$. While the convergence to the minimum is guaranteed, $J(x_i)$ is not guaranteed to converge at the rate $O(1/k)$.

Quality & Clarity:
The manuscript itself is well written and easy to follow. For a further improvement, see my comments in Questions

---

> ### Author Rebuttal · Authors · 2025-07-28
>
> We thank the reviewer for their time and thoughtful feedback. Below we respond to the points raised, highlighting the changes we intend to introduce in the manuscript in response to the comments above.
>
>  - **Originality of MMSE Averaging.** We fully agree that the algorithm is not novel, and we try to clearly state this in the paper, also explicitly linking it to Cold Diffusion (Bansal et al., 2023) and Indi (Delbracio & Milanfar, 2023). We would like to thank the reviewer for pointing out the paper of Feng et al., 2024 which also provides interesting insights connecting iterative denoising to diffusion models. However, our *main contribution is to establish rigorous convergence guarantees* under specific choices of noise levels and averaging weights of the iterative denoising framework. It can be seen as a theoretical justification for many iterative algorithms which have been proposed with empirical and heuristic bases, but with no rigorous justification.
>
>    While the algorithm itself has been widely used before, the fact that the averaging recursion converges towards the proximal operator is not obvious, and requires noting its equivalence with gradient descent on a sequence of smoothed objectives (Prop. 1 in the paper).
>    Likewise, Algorithm 1 is indeed a PnP method and it highlights one of the potential applications which is enabled by having access to a converging approximation to the proximal operator of $\\log p$ (MMSE Averaging). Contrary to many successful PnP methods which approximate the proximal operator without guarantees on its accuracy (e.g. directly replacing it with a denoiser), we provide a theoretical proof that — under our assumptions — the PnP algorithm we propose converges to the MAP estimate.
>  - **Significance under assumptions.** Concerns about the strength of the assumptions are indeed valid. Nevertheless, our goal was to provide a clear and rigorous theoretical foundation in a setting where the analysis is quite delicate, even under strong assumptions. Establishing convergence to the *true* proximal operator and MAP estimator is highly non-trivial and not common in the literature. We believe this is an important first step towards understanding and extending such results to more realistic, potentially non smooth, settings (see also response to Rev. LqTm on the potential extensions to non-convex settings).
>
>    Note that the constraints on the prior distribution do not directly influence the constraints on the network used for MMSE denoising: while ReLU networks are not smooth, they can be used to approximate smooth functions effectively and hence used in our framework; we will clarify this distinction in the paper as it can indeed be confusing.
>
>     To give an example of a more realistic prior which would satisfy our assumptions, we can start with the TV prior, commonly used for images as it penalizes discontinuities in the data. The TV prior for a discrete 1D signal would be $p(x) \\propto \\exp(-\\sum\_{i=1}^{n-1} |x\_{i+1} - x\_i|)$. Since we need the prior to be differentiable, we can use a smooth approximation to the absolute value $|x| \\approx \\sqrt{x^2+\\delta^2}-\\delta$ for small $\\delta$. Then we can calculate the second and third derivatives to verify that the distribution is indeed log concave, and that the bound on the third derivative depends on $\\delta$. This prior is certainly more realistic than the Gaussian prior for image data and is (or was, before deep learning priors emerged) widely used in inverse problems.
>  - **Theorem 2 convergence to the average.** Although last-iterate convergence would be great to have, it typically requires a noticeably more complex analysis. For a good discussion on the difficulties of last-iterate analysis see *"Last iterate convergence of SGD for Least-Squares in the Interpolation regime"*, A. Varre, L. Pillaud-Vivien, N. Flammarion, 2021. In general average-iterate analysis is quite standard in the context of analyzing optimization algorithms, for example see *"Convergence Rates of Inexact Proximal-Gradient Methods for Convex Optimization", M. Schmidt, N. Le Roux, F. Bach, 2011* who prove a similar theorem in a slightly different setting using the average iterate. Note that *best-iterate* convergence can be obtained easily from average iterate convergence. We will highlight this point in the revised manuscript.
>  - **Typo in $\\alpha\_k$.** Thanks for catching this, as you said the correct value is $\\alpha\_k = \\frac{k+1}{k+2}$. Algorithm 1 contained a few inaccuracies, we have restated it in the rebuttal to reviewer 3djj.
>  - **Statement of assumptions.** We will modify the theorem statements to be explicit about the required assumptions in the revised manuscript.

---

> > ### Comment · Reviewer_vJDD · 2025-08-01
> >
> > Thank the authors for their responses. In particular, their claim on the significance is quite convincing. I am going to increase my score.

---

### Official Review · Reviewer_LqTm · 2025-06-30

**Clarity:** 3
**Significance:** 2
**Originality:** 3
**Rating:** 5
**Confidence:** 5

**Summary:**

This paper deals with an algorithm to compute efficiently the proximal operator of some minus log-density $- \ln p$, by using a kind of multi-scale gradient descent that operates on some blurred versions $-\ln p_\sigma$ where $p_\sigma$ is the convolution of $p$ with the density of the Gaussian $\mathcal{N}(0,\sigma^2 I)$.
The proposed method is formulated as an "MMSE averaging" algorithm, which iterates relaxed MMSE denoising with prior $p$.
The authors prove the convergence of this algorithm for a particular choice of step sizes, under the hypothesis that $p$ is positive log-concave and that the third derivative of $\ln p$ is bounded.
They also give an example illustrating the benefit of this algorithm with a quadratic $-\ln p$ corresponding to an ill-conditioned matrix $A$ (in this case, isotropic smoothing improves the conditioning of the matrix and thus the convergence of gradient descent).
The authors also integrates this algorithm in a modified loop (called Approx PGD) in order to target the maximum a posteriori (MAP) of an inverse problem with data-fidelity $f$ and prior $p$. The convergence of this second algorithm is proven under similar hypothese.

**Questions:**

- Could the authors clarify their contribution on Approx PGD compared to existing results about inexact proximal algorithms?

- The proposed method looks quite similar to other algorithms based on simulated annealing. Can the authors compare the proposed algorithm to other ones based on simulated annealing?

- Can the authors confirm that it is really mandatory for their contribution to consider a log-prior $\ln p$, and not just any concave function?

- Are there some interesting non-Gaussian priors on which the proposed results apply? For such a non-Gaussian case, I think that we do not expect the third derivative to be much less than the second one.

**Ethical Concerns:**

["NO or VERY MINOR ethics concerns only"]

**Final Justification:**

After considering the authors' reponses to the reviewers' comments, I increase my score from 4 Borderline Accept to 5 Accept.
This paper brings an interesting contribution to the field of convergence analysis of signal/image restoration algorithms.
The proposed convergence result applies only with a log-concave prior, but seems nevertheless interesting compared to existing results.
Unfortunately, due to this constraining hypothesis, this theoretical result does not apply to recent non log-concave priors that are widely used in the plug-and-play or diffusion literature. But it is likely that the proposed analysis could be extended to a more general setting in the future.

**Limitations:**

yes

**Paper Formatting Concerns:**

No major formatting concern.

**Quality:**

3

**Strengths And Weaknesses:**

# STRENGTHS

- This paper contains interesting thoughts on the way we use MAP or MMSE denoisers in iterative algorithms (like PnP) to solve inverse problems. It makes also a nice connection with recent algorithms based on Cold Diffusion.

- The proposed algorithm is well detailed and relatively easy to implement (except maybe that the number of iterations must increase super-linearly, and that the true MMSE denoiser is rarely tractable).

- The paper is well written with a nice exposition of the main results.

- This paper tackles a very important issue in the literature about PnP and Diffusion, which is to get convergence guarantees of algorithms that operates with a decreasing schedule of noise levels $\sigma_k$.


# WEAKNESSES

- The idea of using an inner loop to approximate the proximal operator in PGD is not new, as shown by the introductive section of the cited paper (Schmidt, Roux,  Bach, 2011) (who quoted several other older references on this specific topic).

- The convergence guarantee applies to an algorithm based on the true MMSE denoiser (for several noise levels $\sigma_k$). In practice, we cannot access exactly these MMSE denoisers, so that another source of error comes at stake. Therefore, the practical benefit of the given convergence bounds seems unclear.

- The convergence guarantees only concern the convex case, but in many applications the most interesting priors are non convex. It appears that the proposed algorithm helps to cope with ill-conditioning of the log prior. In its current form, the paper does not explain how it would be helpful to cope with issues related to non-convexity.

- The only experiment proposed in Section 5 concerns a toy example, which relies on a very well known counter-example of badly conditioned fixed step size gradient descent on a quadratic functional.
 Since a large part of the introduction is devoted to solving inverse problems with plug-and-play or diffusion algorithms, one could have expected to see an experiment showing the benefit of the proposed algorithm in such a setting.
 If the main contribution of the paper is about convergence results on proximal computations, then the paper should give more context about other optimization approaches, and could give less details on the potential applications in imaging.

- The paragraph of the introduction related to PnP methods omits the fact that some of the PnP methods (like the ones of (Cohen et al., 2021a) or (Hurault et al., 2022)) are already formulated with a denoiser that is designed to approximate the MMSE (and not the MAP). And these algorithms already provide convergence guarantees. It would be interesting to know how the proposed algorithm compares with these more direct methods (which do not have an inner loop). Also, it would be interesting to see how the proposed convergence analysis relates to the one given in

> "Provable Convergence of Plug-and-Play Priors with MMSE Denoisers". Xu et al. IEEE Signal Processing Letters, 2020.


## Minor Remarks and Typos

- Saying that $- \ln p$ is ill conditioned in the introduction seems unclear. (In my opinion, it is commonly accepted to talk about an "ill-conditioned matrix" but less common to talk of an "ill-conditioned log-prior").

- It is a bit exaggerated to say that "a new wave of approaches" focus on viewing "inverse problems as a sampling task". This line of reasoning was already present in older papers, see for example

> Louchet, Moisan. "Posterior Expectation of the Total Variation model: Properties and Experiments". SIAM IS, 2013.

- Some of the theoretical results are stated without recalling the hypothese, which make their reading and understanding more difficult. Similarly, when they recall the "classical result by Beck and Teboulle [2009]", they could recall the required hypothesis.

- About the comparison of convergence speed: when $\epsilon \to 0$, the complexity $O(\log(1 / \epsilon))$ is better than $O(1 / \epsilon)$ so that the authors' comment is unclear.

- The step size $\gamma_k$ is not defined when it first appears.

- $J^*$ is not introduced.

- A few sentences are unclear:

> they cannot guarantee that the denoiser is a proximal operator, let alone the proximal operator of the correct functional.

> This assumption controls how skewed and “non-quadratic” the log-prior is

---

> ### Author Rebuttal · Authors · 2025-07-29
>
> We wish to thank the reviewer for their comments and thoughtful engagement with our work. The questions raised in the review in are addressed below.
>
>  - **Compare to existing inexact proximal algorithms.** We of course do not claim novelty in using an inner loop for approximate proximal computations, but we will make sure this is clear in the revised manuscript. As noted by the reviewer, this approach is classical, going back to *Schmidt et al. (2011)* and earlier cited in Section 4. As discussed in lines 299–302, our analysis differs in that the approximation guarantees for the underlying proximal solver are on the *iterates* (from Th. 1), not the objective values. This distinction slightly alters the analysis, particularly in controlling error propagation. Nevertheless, we do not wish to claim this as a major contribution but more as a natural consequence of how our approximation of the proximal operator can be integrated into standard PGD frameworks.
>
>  - **Denoiser approximation error.** Indeed in practice we only have access to approximations of the MMSE denoiser. Our analysis can be extended to take into account this additional source of error: if each time the denoiser is applied, the score estimate is off by $\\varepsilon_k$ (i.e. we get $\\nabla \\log p\_{\\sigma\_k}(x\_k) + \\varepsilon\_k$), and if $|\\varepsilon\_k| \\leq \\varepsilon$ is uniformly bounded, then we can prove convergence to a neighborhood of radius $O(\\varepsilon)$ around the true proximal point at a $1/k$ rate. We will include this extension as a comment in the main text, referring to the appendix for a formal statement.
>
>  - **Dealing with non-convexity.** Our theoretical results rely on the convexity of the negative log prior. Handling the general non-convex setting is outside the scope of our work as it would certainly lead to several additional technical hurdles. One possible way forward is to note that convolution with a Gaussian intuitively "*convexifies*" a probability distribution to some extent. This could significantly help during optimization in practice, and could potentially be useful for theoretical analysis: there exists a minimum $k$ (dependent on the prior $p$) for which $p\_{\\sigma\_k}$ is log-concave. For example, *Stochastic Localization via Iterative Posterior Sampling, L. Grenioux et al., 2024* show in a sampling setting that this *convexification* effect can be used to help sample from non-convex distributions belonging to a restricted class. Another possible direction will be to exploit existing tools allowing to leverage convergence results on convex auxiliary functions, as done by Paquette et al. in *Catalyst Acceleration for Gradient-Based Non-Convex Optimization*. These directions are perfectly relevant, but beyond the scope of our paper. As noted in the discussion with Rev. wDZG, there are of course limitations to our work, but, as far as we know, this is the first convergence result with known rate obtained for such optimization schemes with non-trivial (non-Gaussian) data distributions.
>
>  - **Experiments.** We agree that the toy example in Section 5 is minimal and serves only illustrative purposes, to convey intuition about the conditioning improvements and the trajectory of the iterates. Our primary goal was indeed to provide convergence results and the connections to imaging arised mainly from noting the similarity between MMSE Averaging and PnP algorithms in imaging settings (as well as notably *Cold Diffusion*), where they have shown good empirical performance. We will expand the introduction by adding more context about optimization aspects of the problem. In particular, note that the gradient of (Proximal Objective) is not Lipschitz, hence gradient descent converges as $O(\\frac{1}{\\sqrt{k}})$, and acceleration will not improve this rate. Splitting methods on the other hand will require the prox of $\\log p$, which results in a chicken and egg problem. Finally, higher order methods are also intractable since they will require for example the Hessian of $\\log p$ which is not generally known. Hence the rate of $O(\\frac{1}{k})$ obtained in Theorem 1 is a strict improvement for the specific problem we're dealing with.
>
>  - **Convergence guarantees in PnP.** In the introduction and in the related works section, we noted that there exist many PnP methods with convergence guarantees. However, we also noted that they do not converge towards the MAP solution, but rather to the fixed point of some other operator (different from the proximal operator of $-\\log p$). Our goal is to bridge this gap by showing that one can compute the true MAP minimiser via MMSE denoisers.
>   The analysis in *Provable Convergence of Plug-and-Play Priors with MMSE Denoisers, Xu et al., 2020*, instead follows a different path: the assumptions are much weaker (no convexity required), but convergence is only shown to a stationary point of a function which is not the proximal operator (note the regularizer explicitly stated in Eq. 8 is not equal to $-\\log p(x)$). We will add comparison with this paper to the related work section, since it is indeed relevant.
>
>  - **Is it necessary to work with $\log p$ (i.e., a log-prior), rather than a general concave function?** Yes. Our work requires access to the score function of $-\\log p\_\\sigma$. If instead one starts from a general function $h$, the analogous quantity would be $\\log (\\exp(-h) * \\mathcal{N})$, which is generally intractable as it doesn't have a natural Tweedie-based reformulation. Indeed, as is often the case when dealing with score functions, being able to apply the Tweedie formula is fundamental.
>
>  - **Non-Gaussian priors.** While strong, the assumptions needed for our results to hold do not rule out several interesting non-Gaussian priors. Two examples are
>    1. the smoothed TV prior (where the smoothing can be a simple approximation of the absolute value $|x| \\approx \\sqrt{x^2 + \\delta^2} - \\delta$, or the Pseudo-Huber loss) which can be used as a prior for images penalizing discontinuities in their gradient,
>     2. a uniform distribution over an interval, convolved with a Gaussian to smooth out the discontinuities at the edges.
>
>    Both satisfy Assumptions 1 and 2, and the third derivative is bounded by a quantity which depends on $\\delta$ for 1. and on the standard deviation of the Gaussian in 2. Unlike in the Gaussian case the 3rd derivative is not uniformly lower than the 2nd on the whole of $\\mathbb{R}$. However, note that this does not make the naive GD on the *unsmoothed* proximal objective a good algorithm, because we cannot compute $\\nabla \\log p(x)$, but only $\\nabla \\log p\_\\sigma(x)$ for $\\sigma > 0$.
>
>  - **Comparison to simulated annealing.** Both simulated annealing (SA) and the MMSE Averaging algorithm involve decreasing a "temperature" or noise level which decreases over time to ensure convergence. In our case this serves to deal with non-smoothness of the objective, while in the case of SA it helps deal with non-convexity (or even non-continuity issues). While the principle of reducing the noise is the same, the context of the two algorithms remains quite different: MMSE Averaging is based on gradient descent while SA performs Gibbs sampling at each step to decide the next iterate. Furthermore, while $\\sigma\_k$ in MMSE Averaging does not introduce additional noise, and thus the algorithm is deterministic, SA is inherently a stochastic algorithm. We will add a short comment to clarify the connections with SA.
>
> ### Concerning the minor remarks:
>
>  - We agree that "ill-conditioned log-prior" is not standard and we will clarify this. It is taken from the standard use of "ill-conditioned objective function" from the optimization literature.
>
>  - We will revise the sentence about “a new wave of approaches” and cite earlier works such as *Louchet & Moisan (2013)*, clarifying that this perspective has an older history which has recently gained renewed attention. Thanks for pointing this out.
>
>  - We will restate all theoretical results with full hypotheses for clarity, and similarly recall the assumptions behind the *Beck and Teboulle (2009)* result.
>
>  - Regarding the comparison between $O(\\log(1/\\varepsilon))$ and $O(1/\\varepsilon)$: our point concerns the impact of **conditioning**. We will clarify this in the discussion.
>
>  - We will fix missing notations like $\\gamma_k$ and $J^\\star$, and rewrite unclear sentences accordingly.

---

> > ### Comment · Reviewer_LqTm · 2025-08-06
> >
> > I would like to thank the authors for their precise and thorough answers. I have read all the reviewers' comments and authors' responses. Their responses confirm my interest in this paper, and I am going to increase my score. I still have a few short comments, listed below. Only the last one would deserve an answer, if time permits.
> >
> > **Denoiser approximation error.** It is good that you can prove another convergence result under the assumption that the denoiser can be $\epsilon$-approximated, and I look forward to reading it. However, I would have rather seen this result included in the initial submission so that it could be properly revised. By the way, if you include this result, do not confuse with the notation $\epsilon$ which is already used in the paper.
> >
> > **Dealing with non-convexity.** I understand the fact that the authors decided to put the non-convex case outside the scope of the current paper. Their result appears new and interesting even in the convex case.
> >
> > **Convergence guarantees in PnP.** Indeed, many other convergence guarantees for PnP algorithms prove convergence to a fixed point of some operator, or to a critical point of a non-convex functional. But this is a quite standard limitation when one removes the convexity hypothesis: most algorithms will only go to a critical point or to a local minimum. Targeting the global minimum in the non-convex setting requires more complicated techniques.
> >
> > **Non-Gaussian priors.** I do not completely understand the authors' response on that point. I agree that the smoothed-TV prior is already an interesting non-trivial choice. For the second suggestion ("a uniform distribution over an interval, convolved..."), I am not sure to understand: would this prior be log-concave too?
> > Also, I do not understand the last sentence of the response; is there a typo? For these two priors, I think the situation is the converse one: it is possible to compute explicitly $\nabla \log p(x)$; but $\nabla \log p_\sigma(x)$ is intractable. Therefore, for such priors, naive GD is tractable while MMSE averaging is not. Such cases would already require an approximation of the MMSE denoiser (as discussed in the first point above).

---

> ### Author Response · Authors · 2025-08-07
>
> Thank you for your additional comments and for your positive feedback. Below are some clarifications in response to the points you raised:
>
> **On the approximate denoiser setting**: As is often the case when writing a paper, some natural extensions—while conceptually straightforward—are not initially considered or prioritised. We fully agree that including a convergence result in the presence of denoiser approximation error is relevant. We are happy to provide this result in the revised version and appreciate your suggestion.
>
> **On non-Gaussian priors**:
> After re-reading the corresponding paragraph in our rebuttal, we recognise that it was quite unclear—we apologise for the confusion, and would like to clarify our point here.
>
> The two examples we gave (a smoothed TV prior and the convolution of a uniform distribution with a Gaussian) were meant to illustrate non-Gaussian priors that nonetheless satisfy our assumptions. In particular, the convolution of a uniform distribution with a Gaussian is indeed log-concave due to the preservation of log-concavity under convolution.
>
> However, the final sentence of that paragraph— *“note that this does not make the naive GD on the unsmoothed proximal objective a good algorithm, because we cannot compute $\nabla \ln p$, but only $\nabla \ln p_\sigma$ for $\sigma > 0$”*—was poorly phrased and misleading. What we intended to convey is the following:
>
>    - **While for some toy examples (like the uniform or smoothed TV prior) $\nabla \log p$ may be tractable, in practical settings involving learned denoisers, we do not have access to $\nabla \ln p$, but only to the smoothed scores $\nabla \ln p_\sigma$ for $\sigma > 0$. Therefore, naive gradient descent on the original objective (involving $\log p$) cannot be directly implemented, whereas our proposed method remains applicable.**
>
> We also wish to emphasise that the primary motivation of our work is to address priors that do not have a closed-form expression and are instead learned from data, as is typically the case in modern applications.
>
> We hope this better explains our intent. Please don’t hesitate to reach out if this remains unclear or raises further questions.

---

> > ### Comment · Reviewer_LqTm · 2025-08-08
> >
> > I think the authors for their answer, which solved my last concerns. I will increase my score accordingly.

---

### Official Review · Reviewer_wDZG · 2025-07-02

**Clarity:** 4
**Significance:** 3
**Originality:** 3
**Rating:** 5
**Confidence:** 4

**Summary:**

The authors examine a simple but practical algorithm aimed at solving the MAP optimization objective, $\arg\min_{x \in \mathbb{R}^d} \bigl(\lambda f(x)-\ln p(x)\bigr)$. In general, this optimization objective can be solved using proximal gradient descent, but typically, the proximal operator for $\ln p(x)$ is intractable or unavailable. The authors first propose a simple recursion, called MMSE averaging, that estimates this proximal operator by running gradient descent on a sequence of proximal operators of decreasingly noised distributions. Given some assumptions on log concavity and higher-order derivative boundedness, the authors show this recursion tracks the true proximal operator value. The authors next substitute this recursive proximal calculation for MAP optimization and show $O\left(\frac{1}{k}\right)$ convergence.

**Questions:**

- For learned denoisers (e.g. in diffusion models), could the authors comment on the advantage of using the MMSE averaging algorithm as opposed to directly using access to the parametrized score at a small noise level ($\nabla \log p(x, \sigma=\text{\rm small})$)?

- What is the sensitivity of the convergence guarantees to the specified noise schedule $\sigma_k$ in Theorems 1/2?

- Could the authors comment on the benefit of MMSE averaging relative to using a preconditioner with gradient descent when $\log p(x)$ is tractable?

- What heuristics are there for setting the iteration number in Algorithm 1 for practical implementation?

- Small point, but should $\alpha_k$ be set to $\frac{k+1}{k+2}$ in Proposition 1 and elsewhere?

**Ethical Concerns:**

["NO or VERY MINOR ethics concerns only"]

**Final Justification:**

- The paper is a strong theoretical contribution to a practically relevant problem, and rebuttals from the authors towards points/questions raised confirmed the original positive impression.

**Limitations:**

Limitations are clearly addressed.

**Quality:**

3

**Strengths And Weaknesses:**

Strengths:
- The paper presents definitive theoretical contributions towards a highly relevant "tilted" MAP optimization problem and identifies a simple algorithm to approximate the optimization.

- The explanation of the algorithm as using the noised proximal objectives for an "annealed" or "smoothed" series of gradient descent steps is quite interesting and provides insight towards the compatibility of traditional optimization techniques with pretrained diffusion models.

- Proofs and presentation are very clear and easy to follow.


Weaknesses:
- If $\log p(x)$ is intractable, typically convolving with Gaussian noise will also be intractable, so the algorithm seems to be practically relevant when trained denoisers/score functions are learned, but learned well at the noise levels required.

---

> ### Author Rebuttal · Authors · 2025-07-28
>
> We wish to thank the reviewer for their comments and thoughtful engagement with our work. The questions raised in the review in are addressed below.
>  - **Intractable $\\log p$.** We agree and would like to clarify that, in practice, we do not compute $\\log p\_\\sigma$ directly. Instead, modern denoising models (*e.g.*, trained MMSE denoisers, score-based diffusion models, flow-based models) are explicitly trained to approximate the score $\\nabla \\log p\_\\sigma$ for a range of noise levels. These models can be trained on randomly sampled noise levels, which enables them to interpolate effectively for intermediate values, including those required by our method. Hence, our analysis applies to practical implementations where this score is approximated via learned denoisers.
>  - **Comparison with using $\\nabla \\log p(x, \\sigma = \\mathrm{small})$ directly.** Using a fixed small noise level to approximate the score of the original prior is a valid approach, but as discussed in the paper, it requires performing very small step sizes that can lead to arbitrarily slow convergence to the MAP of a *smoothed* density. In contrast, the scheme we analyze leverages large $\\sigma$ in early iterations and gradually anneals to smaller $\\sigma$ to reach the *true* proximal point with $O(1/k)$ rate (cf. Theorem 1 and paragraph right after, lines 202-218). Importantly, this scheme does not rely on any smoothness parameter about $\\log p$ (*e.g.*, Lipschitz constants), which are typically unknown in practice. There are of course limitations due to the assumptions made about $\\log p$, but, as far as we know, this is the first convergence result with known rate obtained for such optimization schemes with non-trivial (non-Gaussian) data distributions.
>
>  - **Sensitivity to the noise schedule $\\sigma\_k$ in Theorems 1/2.** Our analysis can indeed easily be adapted to a more general class of decreasing noise schedules. In particular, for sequences $\\sigma\_k$ such that $\\sum\_k \\sigma\_k = \\infty$ and corresponding step sizes $\\gamma_k = 1 / (1 + \\tau / \\sigma_k^2)$, we obtain a general convergence rate of the form: $O \\Big( \\min(\\sigma\_k, \\exp( - \\sum\_{j = 1}^k \\sigma_j / \\tau ) \\Big)$ for the smoothed gradient descent recursion of Proposition 1. Thus, any $\\sigma\_k$ decreasing at most as $\\tau / k$ ensures a $O(1/k)$ rate. The specific choice $\\sigma_k^2 = \\tau / (k+1)$ is made to emphasize the connection with the MMSE Averaging scheme $x\_{k+1} = \\alpha\_k \mathrm{MMSE}\_{\\sigma\_k}(x\_k) + (1 - \\alpha\_k) y$, while other choices of noise schedules and averaging weights will lose this equivalent formulation. We will clarify this in the paper.
>
>  - **Comparison to preconditioned GD when $\\log p$ is tractable.** If $\\log p$ and its gradient were tractable, it is very likely that better optimization schemes could be used, but it would fall beyond the typical use case where $p$ represents an unknown data distribution. That being said, theoretical convergence with a preconditioner $P$ would depend on the condition number $\\kappa_P = \\sup\_x \\lambda\_{\\mathrm{max}}(P \\nabla^2 \\log p(x)) / \\inf\_x \ \\lambda\_{\\mathrm{min}}(P \\nabla^2 \\log p(x))$ with a rate of $O(\\kappa\_P \\log(1/\\varepsilon))$. In the best cases this can lead to a considerable speed-up with $\\kappa_P$ close to $1$ but in practice for very badly conditioned functions $\\log p$ the matrix inversion (the optimal $P$ being $(\\nabla^2 \\log p(x))^{-1}$) could lead to severe numerical errors. In contrast, the MMSE averaging scheme does not require any knowledge of $\\log p$ and does not explicitly require any sort of matrix inversion, as long as the MMSE estimator can be computed/well approximated.
>
>  - **Heuristics for the number of iterations in Algorithm 1.** A full empirical study is beyond the scope of our paper, but it is indeed an important future direction to investigate. Existing works such as *Delbracio & Milanfar (2023)* suggest that 5–20 inner denoising steps are often sufficient for high-quality results. We expect our method to behave similarly: the inner loop (MMSE Averaging) benefits from rapid convergence due to improved conditioning in early iterations. We will add a comment to the paper with guidance and references.
>
>  - **$\\alpha\_k$ Correction.** Thank you for catching this. Indeed, there is a typo: the correct choice used throughout the theoretical analysis is $\alpha_k = \\frac{k+1}{k+2}$, not $\frac{1}{k+2}$. We will correct this to be consistent in the paper.

---

> > ### Comment · Reviewer_wDZG · 2025-08-03
> >
> > I thank the authors for their thorough replies and maintain my original (positive) score.

---

### Decision · Program_Chairs · 2025-09-17

**Decision:**

Accept (poster)

**Comment:**

This paper concerns MAP optimization in the case that the proximal operator for the log-prior is unavailable but an exact MMSE denoiser is available.  The authors propose an MMSE averaging recursion that denoises at a sequence of increasingly smaller noise levels, and they prove convergence to the true proximal operator under certain assumption on log-concavity and boundedness of higher-order derivatives.

Four experts reviewed the paper and joined the authors in many rounds of rebuttal and discussion.  Although reviewers LqTm, vJDD, and 3djj all noted that the required assumptions (e.g., log-concavity, thrice differentiability, exact MMSE denoising) limit the applicability of the results in practical settings, the four reviewers all felt that the paper makes a strong theoretical contribution and is well written.  Furthermore, they appreciated the connections to cold diffusion.
Consequently, the paper can be accepted on the condition that the authors incorporate the reviewers' suggestions into the camera-ready version.